# Effective Offline Environment Reconstruction when the Dataset is Collected from Diversified Behavior Policies

## Abstract

In reinforcement learning, it is crucial to have an accurate environment dynamics model to evaluate different policies' value in tasks like offline policy optimization and policy evaluation. However, the learned model is known to have large value gaps when evaluating target policies different from data-collection policies. This issue has hindered the wide adoption of models as various policies are needed for evaluation in these downstream tasks. In this paper, we focus on one of the typical offline environment model learning scenarios where the offline dataset is collected from diversified policies. We utilize an implicit multi-source nature in this scenario and propose an easy-to-implement yet effective algorithm, policy-conditioned model (PCM) learning, for accurate model learning. PCM is a meta-dynamics model that is trained to be aware of the evaluation policies and on-the-fly adjust the model to match the evaluation policies' state-action distribution to improve the prediction accuracy. We give a theoretical analysis and experimental evidence to demonstrate the feasibility of reducing value gaps by adapting the dynamics model under different policies. Experiment results show that PCM outperforms the existing SOTA off-policy evaluation methods in the DOPE benchmark with *a large margin*, and derives significantly better policies in offline policy selection and model predictive control compared with the standard model learning method.

## 1 Introduction

Environment model learning, which learns a model to approximate state transitions and reward functions of the environment, has extensive applications in Offline Policy Evaluation (OPE) (Thomas et al., 2015; Doroudi et al., 2017) and offline Reinforcement Learning (offline RL) (Lange et al., 2012; Levine et al., 2020). In OPE, policy value is estimated by calculating the return of simulated trajectories from the learned model. In offline RL, approaches utilize the model for planning or optimizing policy to maximize the return. Model accuracy significantly affects the efficacy of these methodologies. However, a learned model is known to have a large value gap when used to evaluate a target policy different from the data-collection policies (Xu et al., 2020; Clavera et al., 2018; Janner et al., 2019; Yu et al., 2020). This issue has hindered the adoption of models in many scenarios.

In this article, we focus on learning an accurate dynamics model for offline policy optimization and policy evaluation. Most current model learning methods fit the whole dataset with a unified model and then utilize it for evaluation no matter what target policy is faced with. We refer to it as *Policy-Agnostic dynamics Model (PAM)*. In many realistic situations such as robotic manipulation (Mandlekar et al., 2018; 2021), autonomous driving (Yu et al., 2018; Sun et al., 2020) and sequential recommendation (Saito et al., 2020; Gao et al., 2022), the offline data is collected by a wide range of different policies (including parameterized policies, rule-based policies and also human policies), and this inherent multi-source nature of the dataset remains rarely explored in recent works on dynamics model learning. In these learning tasks where the diverse data-collection behavior policies in fact correspond to different sources of state-action distributions (Ho & Ermon, 2016), the distribution of the offline dataset will be broad. Due to the state-action visitation frequency shift among different policies, state-action pairs visited by one policy may be infrequently observed by another policy, resulting in the offline data not always being beneficial for accurately evaluating the current policy. As shown later, attempting to learn samples from all policies may even impair the accuracy of predictions for the current policy. From this perspective, model learning from offline datasets collected by numerous behavior policies implies a feature of data fitting from *a mixture of multi-source distributions, which is ignored in current learning paradigms.*

For accurate model learning in this scenario, we utilize the implicit multi-source nature caused by numerous data-collection policies and propose an easy-to-implement yet effective algorithm, policy-conditioned model (PCM) learning. PCM is a meta-dynamics model that is trained to be aware of the evaluation policies and make predictions by adapting to the evaluation policies' state-action distribution to improve the prediction accuracy. In practice, we implement PCM via policy representation techniques (Duan et al., 2016; Chen et al., 2021; Nagabandi et al., 2019), which adopt an extra policy-aware module to on-the-fly encode policies' representation and input the policy representations as well as a state-action pair into the meta-dynamics model. PCM produces different dynamics models given different policy representations. We theoretically show that PCM can achieve a smaller value gap for a target policy compared with PAM.

Experiments are conducted based on MuJoCo (Todorov et al., 2012). We first conducted a proof-of-concept experiment, utilizing our custom-made dataset, which verified the effectiveness of the policy-aware mechanism for improving the model prediction accuracy. Then apply PCM in several downstream tasks. Results show that PCM improves the performance of off-policy evaluation in the DOPE benchmark with *a large margin*, and derives significantly better policies in offline policy selection and model predictive control compared with the standard model learning method.

## 2 PRELIMINARIES

### 2.1 MARKOV DECISION PROCESS AND REINFORCEMENT LEARNING

We consider a Markov decision process (MDP) (Sutton & Barto, 2018) specified by the tuple $\mathcal{M} = (\mathcal{S}, \mathcal{A}, r, T, \gamma, \rho_0)$, where $\mathcal{S}$ is the state space, $\mathcal{A}$ is the action space, $r(s, a)$ is the reward function, $T(s'|s, a)$ is the transition function, $\gamma \in (0, 1)$ is the discount factor, and $\rho_0(s)$ is the initial state distribution. In reinforcement learning (RL), we are typically concerned with optimizing or estimating the value of a policy $\pi$ in a policy space $\Pi$. Specifically, value is defined as:

$$V^\pi = \mathbb{E}_{s_0 \sim \rho_0, s_{1:\infty}, a_{0:\infty} \sim \pi} \left[ \sum_{t=0}^{\infty} \gamma^t r(s_t, a_t) \right]. \tag{1}$$

For a fixed policy $\pi$, the MDP becomes a Markow chain, and we define the occupancy measure $\rho^\pi(s, a) = (1 - \gamma) \sum_{t=0}^{\infty} \gamma^t \mathbb{P}_\pi(s_t = s, a_t = a)$ then the policy value can be rewritten as $V^\pi = \mathbb{E}_{s, a \sim \rho_{T*}^\pi}[r(s, a)]$. When different dynamics are involved, we use an additional subscript to indicate the transition, e.g. $V_{T*}^\pi$ and $V_{\hat{T}}^\pi$.

### 2.2 OFF-POLICY EVALUATION

Off-policy evaluation (OPE) (Le et al., 2019; Precup et al., 2000; Jiang & Li, 2016; Kostrikov & Nachum, 2020; Yang et al., 2020; Wen et al., 2020) aims at estimating the value $V^\pi$ of a target policy $\pi$, based on a fixed dataset of transitions $\mathcal{D}$ collected from some *behavior policies* $\{\mu_i\}_{i=1}^n$ (or named *data-collection policies*). This problem is of great practical significance for several reasons, including providing high-confidence guarantees prior to deployment, performing policy improvement, and model selection. A major challenge in OPE is the distribution shift between the behavior policy and the target policy, which induces a large value gap between the estimated value and the true value.

## 3 RELATED WORKS

**Off-policy Evaluation (OPE):** OPE research is relevant to many practical domains such as recommendation systems (Li et al., 2011), health (Liao et al., 2019), and education (Mandel et al., 2014). There exists a large body of work on OPE, including methods based on fitted q-evaluation (Le et al., 2019; Hao et al., 2021) and importance sampling (Kostrikov & Nachum, 2020). Another class of OPE is the model-based approach (also referred to as the direct method), which is focused on in this paper. While model-based OPE has been considered by many previous works (Thomas & Brunskill, 2016; Hanna et al., 2017), they are confined to simple tasks and produce biased predictions owing to the restricted range of state and action space in offline trajectories (Fu et al., 2021b). . By contrast, our approach is applied to more intricate tasks and proves that model-based OPE can also do well in challenging continuous tasks.

**Model should be aware of policies:** There are some previous works in other fields also proposing the idea that the dynamics model should be aware of or focus on certain policies rather than all the policies. PAML (Abachi et al., 2020) proposes that model learning should incorporate the way the planner is going to use the model. PDML (Wang et al., 2022) dynamically adjusts the historical policy mixture distribution to ensure the learned model can continually adapt to the state-action visitation distribution of the evolving policy. However, in contrast to us, both of them are concerned with the

online RL setting and achieving the policy-aware mechanism by adjusting the sampling distribution from the replay buffer. Our method considers the offline RL setting and explicitly incorporates policy representation as an extra input for model learning.

**Model-based Offline RL:** Model-based Offline RL (MBORL) algorithms also involve dynamics models for some downstream tasks. From the perspective of model usage, MBORL can generally be categorized into two groups: model predictive control (MPC) (Camacho & Alba, 2013) and Policy learning (PL). In MPC, Argenson & Dulac-Arnold (2021) directly performs planning in a learned dynamics model. In PL, a policy can be trained either in an in-support region by utilizing a conservative surrogate MDP (Yu et al., 2020; Kidambi et al., 2020; Yu et al., 2021), or in out-of-policy regions by learning an adaptive policy (Chen et al., 2021). Some works also utilize dynamics models with off-the-shelf model-free algorithms for better policy learning (Lyu et al., 2022; Wang et al., 2021). Recent studies (Rigter et al., 2022; Yang et al., 2022) also adopt an adversarial framework that alternates between dynamic-model training and policy learning. However, these works pay more attention to optimizing policy under a restricted dynamics model instead of directly learning a faithful model when using it, where the latter is what our work focuses on.

## 4   POLICY-CONDITIONED DYNAMICS MODEL LEARNING

In this section, we first give the metric to evaluate the gap between true dynamics and a learned model in Sec. 4.1 and the intuition for policy-conditioned model (Sec. 4.2). Then we formally introduce the policy adaptation mechanism of PCM from an error reduction perspective (Sec. 4.3) and show this mechanism also leads to a better generalization to out-of-distribution data (Sec. 4.4).

### 4.1   VALUE GAPS BETWEEN TRUE DYNAMICS AND A LEARNED MODEL

Offline dataset $\mathcal{D} = \{\tau_m\}_{m=1}^M$ consists of previously collected trajectories $\tau_m = (s_0, a_0, r_0, s_1, \dots)$, each of which is generated by the interaction between one of the behavior policies $\Omega = \{\mu_i | i \in \mathcal{I}\}$ and the environment. Here we consider the multiple diversified behavior policy case, which coincides with many realistic situations. It should be noted that this is not a setting raising new challenges but a refined description of the existing problem, which provides more information that could be utilized for dynamics modeling compared to simply ignoring the multi-source property of the dataset.

We follow the basic idea in OPE to define the performance metric of a dynamics model: in an MDP, a good dynamics model means for any target policy $\pi$, the gap between the value under true transition $T^*$ and the value estimation under $\hat{T}$ is small, i.e., $|V_{T^*}^\pi - V_{\hat{T}}^\pi|$ is small. Inspired by a previous study (Janner et al., 2019), the value gaps between true dynamics and a learned model is bounded by

$$|V_{T^*}^\pi - V_{\hat{T}}^\pi| \leq \frac{2R_{\max}\gamma}{(1-\gamma)^2} l(\pi, T^*, \hat{T}), \tag{2}$$

where $l(\pi, T^*, \hat{T}) = \mathbb{E}_{s,a \sim \rho^\pi} D_{\mathrm{TV}}(T^*(\cdot|s,a), \hat{T}(\cdot|s,a))$ is total variation divergence between true and learned transitions under the state-action occupancy of the target policy $\pi$ to measure the model error. Eq. (2) implies that as long as we reduce the model error $l(\pi, T^*, \hat{T})$ under the target policy's distribution $\rho^\pi$, we can guarantee the reduction of the corresponding upper bound of the value gap. The bound is an extension of previous bounds in Janner et al. (2019); Xu et al. (2020; 2021), where we further consider the generalization ability of the learned models. The full derivation is in App. A.1.

### 4.2   THE INTUITION FOR POLICY-CONDITIONED MODEL LEARNING

In Fig. 1, we use an example to illustrate why policy-conditioned model (PCM) learning is superior to policy-agnostic model (PAM) learning.

Suppose we wish to learn an environment model where a biped robot is asked to move forward from an offline dataset including different locomotion patterns, such as walking, running, jumping, etc. Currently, the standard dynamics model, i.e., the policy-agnostic model (PAM), learns to predict all of the transitions coming from different locomotion patterns in one unified model. However, we notice that different locomotion patterns usually correspond to quite different transition patterns though these patterns can be regarded as a single task. For instance, jumping requires both legs to be folded and unfolded at the same time while running involves alternate flexion and extension of the legs. If we can utilize this nature, the learning complexity will be reduced.

Based on the above motivation, instead of learning a single model for the whole dataset, we propose to "divide" the dataset according to the data-collection policy and learn a model for each subset.

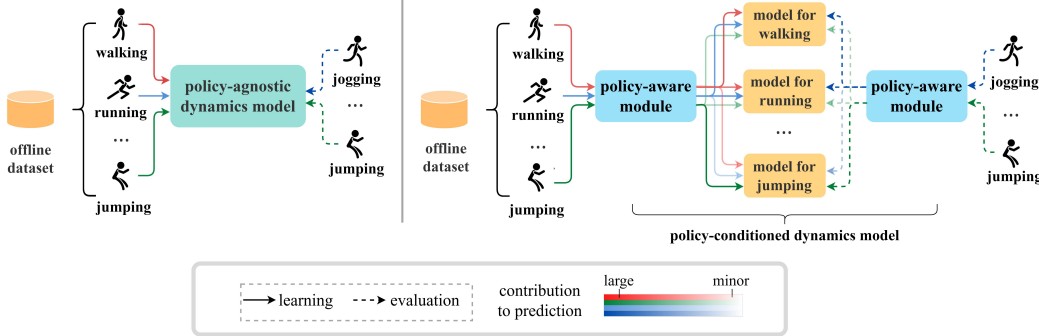

Figure 1: An illustration of the difference between the policy-agnostic model (left) and the policy-conditioned model (right). Suppose we wish to learn an environment where a biped robot is asked to move forward from an offline dataset including different locomotion patterns, such as jumping, walking, running, etc. Different locomotion patterns usually correspond to quite different transition patterns even though they can be regarded as a single task.

We regard each locomotion pattern as a subtask and respectively learn a model for each subtask. In this way, we can reduce the learning difficulty of each model, which is expected to obtain a more accurate model for each data-collection policy. The rationale behind this is that each data-collection policy only focuses on a relatively small subregion of the support set of the whole mixed state-action distribution, thus training the model under the state-action occupancy of each policy should be an easier task than the global model training and tends to obtain more accurate models. Moreover, if the target policy to be evaluated is unseen before in the dataset, e.g. jogging, which is a locomotion pattern between walking and running, it is hoped to yield a new model to adapt to the jogging policy by combining the walking model and the running model.

### 4.3 THE POLICY ADAPTATION MECHANISM FOR MODEL LEARNING

With a dataset $D$ collected by a set of diversified behavior policies $\Omega = \{\mu_i | i \in \mathcal{I}\}$, the training data distribution is a mixture of occupancy measures $\rho^{\text{mix}}(s, a) = \sum_{i \in \mathcal{I}} w_i \rho^{\mu_i}(s, a)$, where $w_i$ is data proportion of policy $\mu_i$. Conventional model learning fits a universal transition model directly under the whole mixed data distribution and rolls out whatever target policy in this *policy-agnostic model*:

$$\hat{\psi} = \arg \min_{\psi \in \Psi} \sum_{\mu_i \in \Omega} w_i l(\mu_i, T^*, T_\psi), \tag{3}$$

where model $T$ is parameterized by $\psi \in \Psi$. This is sufficient for simple environments where the model capacity is rich enough to completely recover true transitions. However, in realistic large-scale tasks, the model's capacity is limited in comparison to true transition, resulting in a non-zero error, which will be further compounded during long-horizon rollout (Janner et al., 2019; Xu et al., 2020).

With an adequate model capacity, it is possible to accurately fit true transition dynamics, which is the *unique* optimal model for any target policy. Nevertheless, the usually limited model capacity prevents perfect transition modeling and requires a proper allocation of the finite accuracy budget to facilitate the target policy rollout as much as possible. Since different policies perform distinct behaviors and access varied subregions of the state-action space, their optimal models within the model space are different, resulting in an optimal model *inconsistency*, i.e., there does not exist a unique model within the model space that is optimal for general target policies.

A consequent idea is to select dynamics models adaptively for different policies, where each model is optimized specially for the occupancy measure of its corresponding policy. We name it *policy-conditioned model (PCM)*. This "model selection" procedure can be expressed through a mapping $F : \Pi \to \Psi$, where each policy $\pi$ is associated with a model $T_{F(\pi)}$. Learning a PCM is therefore translated into finding an optimal $F$ to minimize model error on the data distribution of each policy:

$$\hat{F} = \arg \min_{F \in \mathcal{F}} \sum_{\mu_i \in \Omega} w_i l(\mu_i, T^*, T_{F(\mu_i)}), \tag{4}$$

where $\mathcal{F}$ is function space of $F$. For behavior policies $\mu_i$, model error $l(\pi, T^*, T_{\hat{F}(\mu_i)})$ can be reduced to achieve smaller value gaps compared to PAM as shown empirically in the experiments in Sec. 5.1 and 5.2. This is intuitive since PAM attempts to fit global transition dynamics, which is more difficult than local transition modeling that PCM specializes in. For those new target policies $\pi$, the learned models have to extrapolate to the data distribution $\rho^\pi$, resulting in an extra generalization error. We show a generalization benefit brought by the adaptation mechanism in the next section.

**Remark 1 (Varied dynamics models for different policies):** The formulation of PCM is similar to a meta-learning objective with $F$ representing the meta module (Rakelly et al., 2019). At first glance, it is counterintuitive to build the problem as a meta-optimization problem since all policies are deployed in the same environment $T^*$, meaning that the ground-truth parameter of the dynamics model $T_\psi$ among different policies $\pi$ *should be the same*, while the PCM gives varied models adapted to different policies. In fact, this adaptation method under limited model capacity resembles human behaviors under limited attention. For example, when a man drives a car, his attention focuses on the road, and therefore the predictions of vehicle movement are relatively clear in his mind while the flight trajectories of the birds in the sky are blurred. On the contrary, when the man stops the car and starts to observe the flying birds, his focus will be shifted. Even in the same environment, the models in his brain differ in the accuracy assignment when performing different tasks. This adaptability reflects an attempt to maximize efficiency in the use of limited attention (or capacity), and our policy-conditioned model actually shares a common idea.

**Implementation:** In real-world applications, the corresponding white-box policies are typically unknown. It is impractical to learn a mapping function $F(\pi)$ which directly takes policy $\pi$ as the input. Inspired by many previous works (Duan et al., 2016; Chen et al., 2021; Nagabandi et al., 2019) which have successfully utilized RNN as an extra representation extractor module to map the interaction trajectories to some task-specific meta-parameters, we use similar RNN structure to learn and infer policy representations from given interaction trajectories, and a **policy-representation-conditioned dynamics model** is learned to adapt its predictions based on the input policy representation. Formally, let $\tau_{0:t} = (s_0, a_0, s_1, a_1, ..., s_t, a_t)$ be a trajectory generated by a data-collection policy up to timestep $0 \leq t \leq H - 1$ ($H$ is the horizon of the MDP) and the offline dataset is a set of $N$ trajectories $\mathcal{D} = \{\tau^{(j)}\}_{j=1}^N$. For any timestep $t$, trajectories $\tau_{0:t-1}$ will be fed into a recurrent neural network $q_\phi(\tau_{0:t-1})$ to obtain an embedding $z_t$. After that, an adaptive dynamics model $T_\psi(s_{t+1}|s_t, a_t, z_t)$ is learned to adapt its predictions of $s_{t+1}$ based on $z_t$. Recall that we expect to get a representation of a policy, the embedding $z_t$ should encode salient information about the policy. To this end, we simply incorporate a policy decoder $p_\theta(a_t|s_t, z_t)$ and the encoder and decoder are jointly optimized to reconstruct the specified policy. In summary, the overall learning objective of PCM is:

$$\min_{\phi, \theta, \psi} \mathbb{E}_{t \sim [0, H-2], \tau_{0:t+1} \sim \mathcal{D}} [- \underbrace{\log T_\psi(s_{t+1}|s_t, a_t, q_\phi(\tau_{0:t-1}))}_{\text{adaptive model learning}} - \lambda \underbrace{\log p_\theta(a_t|s_t, q_\phi(\tau_{0:t-1}))}_{\text{policy representation}}], \quad (5)$$

where $\lambda$ is a hyperparameter. Note that the gradients would be backpropagated from $T_\psi$ and $p_\theta$ to $z$ if optimal models' or policies' parameters in different trajectories are inconsistent but have the same representation of $z$, then the parameters of $\phi$ will be updated automatically to distinguish them. Our pseudo-code of the overall PCM learning is shown in Alg. 1.

**Remark 2 (Model complexity):** The module $F$ in PCM (i.e. the RNN module in our implementation) introduces additional model complexity compared to the model space of PAM. One may suspect that it is the increased model capacity helps the dynamics model learning, instead of the policy-conditioned mechanism. In fact, we find that simply increasing the model capacity by using a larger network without mechanism changes cannot bring significant improvement (as shown in Sec. 5.3.1). The additional module works mainly because it allows an adaptive and therefore effective utilization of the limited capacity for different target policies, which reduces the in-distribution model error and also brings generalization benefits for new target policies as we show in the next subsection.

## 4.4 ADAPTATION EFFECT IMPROVES THE GENERALIZATION

In this section, we show that the adaptation effect from PCM can provide additional generalization benefits when the learned model extrapolates to the data distribution of new target policies absent from the training dataset. We introduce an assumption on the smoothness of well-trained models:

**Assumption 4.1.** For the learned model $T$, the point-wise model error $D_{\text{TV}}(T^*(\cdot|s, a), T(\cdot|s, a))$ is $L$-Lipschitz with respect to the state-action pairs, i.e.,

$$\left| D_{\text{TV}}(T^*, T)(s_1, a_1) - D_{\text{TV}}(T^*, T)(s_2, a_2) \right| \leq L \cdot D\big((s_1, a_1), (s_2, a_2)\big), \quad (6)$$

where $D(\cdot, \cdot)$ is some kind of distance defined on the state-action space $\mathcal{S} \times \mathcal{A}$.

Assump. 4.1 measures the local generalization ability of a learned model. Generally speaking, if we say the learned model $T_{\hat{\psi}}$ generalizes well w.r.t. the state-action inputs, we mean that for some unseen $(s_2, a_2)$ deviating from a training data $(s_1, a_1)$, the point-wise model error will not increase much, reflected by a bounded $L$. Based on this assumption, we find that the expected model error of PCM under the target policy data distribution can be controlled:

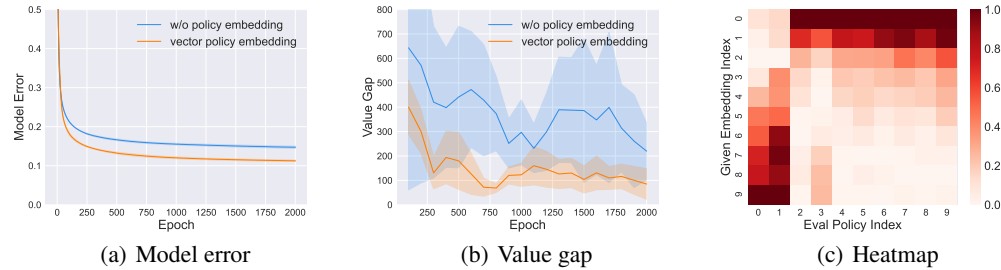

| (a) Model error | (b) Value gap | (c) Heatmap |

Figure 2: Illustration of model error, value gap of policy learned in with or w/o policy embedding model, and the heatmap about the performance of evaluating policy with different policy embedding.

**Proposition 4.2.** *Under Assump. 4.1, for any policy $\pi \in \Pi$, model error of PCM $T_{\hat{F}(\pi)}$ is bounded:*

$$l(\pi, T^*, T_{\hat{F}(\pi)}) \leq \min_{\mu_i \in \Omega} \left\{ \underbrace{l(\mu_i, T^*, T_{\hat{F}(\mu_i)})}_{\text{training error}} + \underbrace{L \cdot W_1(\rho^\pi, \rho^{\mu_i}) - C(\pi, \mu_i)}_{\text{generalization error}} \right\}, \tag{7}$$

*where adaptation gain $C(\pi, \mu_i) := l(\pi, T^*, T_{\hat{F}(\mu_i)}) - l(\pi, T^*, T_{\hat{F}(\pi)})$, $W_1$ is Wasserstein-1 metric.*

The adaptation gain $C(\pi, \mu_i)$ summarizes the benefit of the policy adaptation effect based on the insight that when testing on a new policy $\pi$ within some effective region, the model $T_{\hat{F}(\pi)}$ customized for $\pi$ should have a smaller model error under the target distribution $\rho^\pi$ than any $T_{\hat{F}(\mu_i)}$. PAM does not include the policy-conditioned mechanism and the adaptation gain is always zero. Simply fine-tuning the PAM parameters for a new policy is not practical because it requires the interaction of the new policy with the environment to collect the target domain experiences, which is prohibitive in general. In contrast, the policy representation serves as an additional covariate in PCM, which enables an extra adaptation ability to target policies and hence a non-zero adaptation gain term with no need for the real experiences in the target domain and also the model parameter fine-tuning. Therefore, Prop. 4.2 shows that the model error of PCM for a new target policy $\pi$ is reduced by the adaptation gain $C$, if $C > 0$, compared with PAM.

However, it is hard in general to rigorously analyze the adaptation gain $C(\pi, \mu_i)$ because of the complexity of neural networks and the optimization process. Empirically, as the target policy $\pi$ gradually diverges from $\Omega$, the adaptation gain will increase from zero and partially reduce the extrapolation error within an effective adaptation region. When $\pi$ leaves far enough from $\Omega$, $C$ will reach the maximum and then start to decrease. This trend exhibits the efficacy of policy adaptation to a reasonable degree. We provide experimental evidence in Sec. 5.2, which aligns with the intuition. We also discuss two extreme cases of zero adaptation effect and complete cancellation of extrapolation error in App. A.3, and the realistic case lives between the two extremes.

## 5 EXPERIMENT

In this section, we first justify the efficacy of the policy adaptation mechanism for model learning via a proof-of-concept experiment (Sec. 5.1). In Sec 5.2, we conduct experimental studies to verify PCM enjoys smaller value gaps as analyzed in Sec. 4.4. Then we evaluate PCM on specific downstream tasks including off-policy evaluation (OPE), offline policy selection (OPS) and model predictive control (MPC), in contrast to PAM (Sec 5.3). Finally, we analyze the learned policy embedding by PCM to verify whether it learns reasonable policy representation (Sec 5.4) [1].

### 5.1 PROOF-OF-CONCEPT VERIFICATION ON THE POLICY ADAPTATION MECHANISM

We consider a simplified setting that does not involve generalization to unseen policies to justify the idea of the policy adaptation mechanism for model learning. We collect a dataset sampled by 10 different policies in HalfCheetah and solely choose one of the 10 policies for evaluation. Since there is no need for generalization, we can use a simple policy representation scheme called *vector policy embedding*, $F(\mu_i)$. Specifically, we employ a $n \times m$ matrix to represent the policies, where $n$ is the number of policies in the dataset and $m$ is the dimension of the policy representation. The matrix can be updated by backpropagation. We compared the performance of the model with and without embedding. Fig. 2(a) and 2(b) show even with such a simple policy representation scheme, PCM can significantly outperform PAM on the model error as well as the value gap.

Furthermore, we show that the vector policy embedding indeed helps the model adapt to a specific policy. We first train and obtain an embedding for each policy, After training, we have 10 different

---

[1]code: the code will be published after the acceptance.

vector policy embeddings for these 10 policies, respectively. Then we evaluate each policy under models given different vector embeddings and record the value gap under each case. The results are shown in the mismatch heatmap below. Fig. 2(c) shows that the model performs better under policy with better-matched embedding (for any two policies, the closer their numbers are, the more similar they are), indicating that the vector policy embedding helps the model adapt to a specific policy.

## 5.2 EMPIRICAL EVIDENCE OF PCM HAVING SMALLER VALUE GAPS

Prop. 4.2 indicates that the value gap of PCM for an unseen policy $\pi$ can be reduced by *1) a smaller model error on the training dataset; 2) a positive adaptation gain C*. We now present empirical evidence to support our analysis and demonstrate that PCM indeed has smaller value gaps. All experiments in this section are conducted in the HalfCheetah environment.

We first compare the model error of PAM and PCM on the training dataset. As shown in Fig. 3(a), PCM enjoys a smaller model error than PAM. We then analyze the adaptation gain quantitatively by fixing a data-collection policy $\mu_i$ and computing $C(\pi, \mu_i)$ for different policies $\pi$. We refer to Appx. E.2 for more details. As illustrated in Fig. 3(b), the gain gradually increases with the policy divergence, reaches a maximum, and decreases as the policy divergence continues increasing. This confirms the analyzed cases in Sec. 4.4.

Finally, we directly compare the value gaps of PAM and PCM and also investigate the influence of different levels of dataset diversity on them. To do so, we construct datasets with varying levels of diversity (0%, 20%, 50%, 80%, 100%), where the percentage indicates that the dataset is created from the replay buffer of SAC (Haarnoja et al., 2018) until the policy reaches the specific level of performance. Appx. E.1 presents details of the data collection process. We train PAM and PCM on each dataset and test them on other 11 policies provided by the DOPE benchmark (Fu et al., 2021a), which were unseen before in the datasets. Fig. 3(c) depicts the value gap of each model trained on each dataset, demonstrating that PCM can achieve smaller value gaps. Moreover, the results show that as the diversity of the dataset increases, both PAM and PCM achieve smaller value gaps, with PCM exhibiting a more substantial advantage.

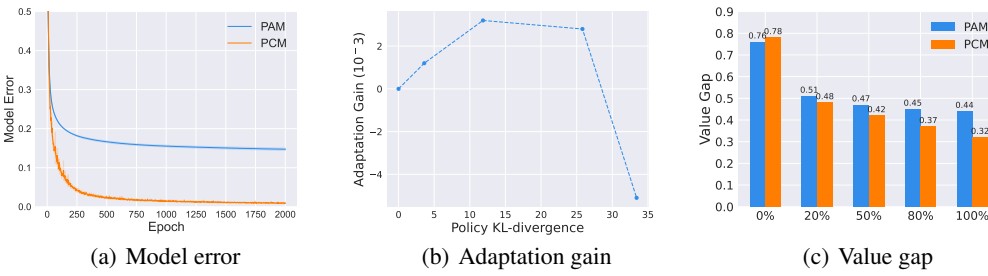

(a) Model error       (b) Adaptation gain       (c) Value gap

Figure 3: Left illustrates model errors (computed by mean squared error) of PAM and PCM on the training dataset. Medium illustrates the adaptation gain of PCM for different unseen policies $\pi$, relative to a data-collection policy $\mu_i$. The right illustrates normalized value gaps of PAM and PCM trained on datasets with different levels of diversity when testing on 11 target unseen policies.

## 5.3 EVALUATION ON DOWNSTREAM TASKS

### 5.3.1 OFF-POLICY EVALUATION

We compare PCM with several OPE methods, including: **Fitted Q-Evaluation (FQE)** (Le et al., 2019), that estimates the policy value via iteratively performing Bellman update, **Doubly Robust (DR)** (Jiang & Li, 2016), that combines the importance sampling technique with a value estimator for variance reduction, **Importance Sampling (IS)** (Kostrikov & Nachum, 2020), that performs importance sampling with a learned behavior policy, **DICE** (Yang et al., 2020), that uses a saddle-point objective to estimate marginalized importance weights $d^\pi(s,a)/d^{\pi_B}(s,a)$, **Variational Power Method (VPM)** (Wen et al., 2020), that runs a variational power iteration algorithm to estimate the importance weights without the knowledge of the behavior policy, **Policy-Agnostic Model (PAM)**, that removes the policy representation module in PCM and serves as the ablation method. We evaluate these approaches on a variety of tasks from DOPE-D4RL and DOPE-RL-Unplugged benchmarks (Fu et al., 2021a). The data in these tasks is collected by diverse policies, which aligns with the multi-source assumption in our theoretical analysis.

Fig. 4 shows the performance of PCM and other methods in three metrics (details of the metrics and results separated by tasks are in App. C). We find that PCM outperforms other methods by a large

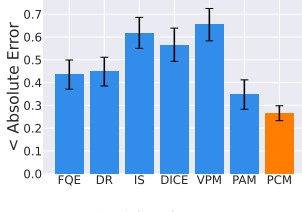
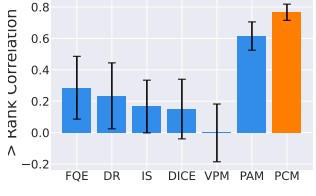
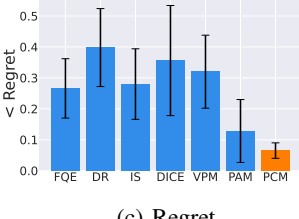

| (a) Absolute error | (b) Rank correlation | (c) Regret |
|---|---|---|

Figure 4: The overall performance of OPE in three metrics. To aggregate across tasks, we normalize the real policy values and evaluated policy values to range between 0 and 1.

margin. Specifically, the results of the absolute error provide direct evidence that PCM can reduce the value gap effectively. Besides, PCM obtains a higher rank correlation and lower regret, indicating that PCM can not only perform accurate evaluation but also select the competitive policies among the policies to be evaluated.

Note that PAM also shows competitive performance among these algorithms, which contradicts results from most previous works (Fu et al., 2021a; Voloshin et al., 2019). This is because we incorporate some components of modern neural networks both into PAM and PCM. To be more specific, we find that classical MLP (denoted as PAM(old arch)) is not well-suited for autoregressive predictions when evaluating a policy and is susceptible to compounding errors, as shown in Fig. 5. After introducing components of modern neural networks, including residual connection, layer normalization, and dropout, into our baseline (denoted as PAM(new arch)), we observe a significant reduction in the compounding error as well as a remarkable improvement in overall performance, as illustrated in Fig. 5 and Tab. 1.

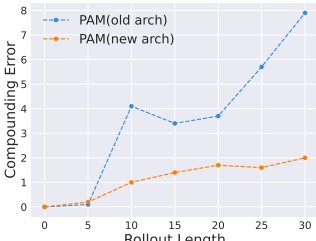

Figure 5: Illustration of compounding error of different network architectures for PAM.

Table 1: OPE performance of different networks, in terms of absolute error (value gap), rank correlation, and regret. We **bold** the best scores for each metric.

| Method | Absolute error | Rank correlation | Regret |
|---|---|---|---|
| PAM (old arch) | 0.51±0.04 | 0.47±0.10 | 0.22±0.07 |
| PAM (new arch) | 0.29±0.06 | 0.62±0.09 | 0.13±0.10 |
| PAM (new arch larger) | 0.26±0.05 | 0.61±0.07 | 0.12±0.08 |
| PCM | **0.19±0.03** | **0.77±0.05** | **0.07±0.03** |

Furthermore, to keep a balanced capacity, we increase the size of the network for PAM (from 200 hidden size & 4 layers to 400 hidden size & 4 layers (denoted as PAM(larger))) and the result is shown in Tab. 1. It shows that even with increasing the size of PAM, PAM still falls behind PCM.

### 5.3.2 OFFLINE POLICY SELECTION

In this section, we explore the efficacy of using PCM on offline policy selection (OPS) for a practical offline RL algorithm. Specifically, we train MOPO (Yu et al., 2020) for 1000 epochs and record policy snapshots at the latest 20 epochs for OPS. We compare our method against PAM and FQE as well as directly selecting the last-epoch policy. Tab. 2 shows the performance gains by different methods. The performance gain is computed by $\frac{(V_{\text{selected}} - \bar{V})}{V_{\text{max}} - \bar{V}} \times 100\%$, where $V_{\text{selected}}$ is the value of the selected policy and $\bar{V}$, $V_{\text{max}}$ are the average and max values of the evaluated policies, respectively. It is noteworthy that the gains of FQE and PAM are even lower than directly selecting the last-epoch policy, also indicated in another work (Qin et al., 2022). In contrast, our approach shows a brilliant performance, implying that it reliably chooses a better policy for an offline RL algorithm to deploy.

Table 2: Performance gain of offline policy selection for MOPO (Yu et al., 2020) by different methods.

| Task Name | Last Epoch | FQE | IS | DICE | PAM | PCM (Ours) |
|---|---|---|---|---|---|---|
| halfcheetah-medium-replay | 39.3% | 23.0% | 87.8% | 1.6% | 1.6% | **98.4%** |
| hopper-medium-replay | 56.0% | 34.1% | 56.0% | 19.8% | 47.3% | **64.8%** |
| walker2d-medium-replay | -4.6% | 4.6% | 34.3% | 13.0% | -30.6% | **51.9%** |
| Average | 30.2% | 20.6% | 59.4% | 39.3% | 11.5% | **71.7%** |

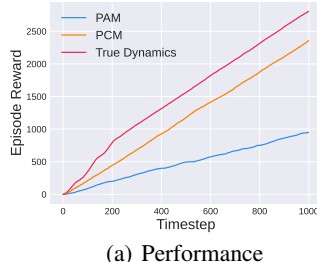 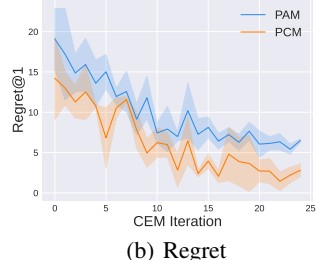

|             (a) Performance             |               (b) Regret               |

Figure 6: Left illustrates the cumulative rewards within an episode in HalfCheetah task. Right illustrates the regrets of PAM and PCM during CEM, obtained by tracking several planning processes.

### 5.3.3 MODEL PREDICTIVE CONTROL

An accurate model can also be expected to perform effective model predictive control (MPC). We therefore compare our proposed PCM against PAM and the true dynamics (using MuJoCo simulator itself as true dynamics). Following Chua et al. (2018), we use the cross-entropy method (CEM) as the optimization technique in MPC, which samples actions form a distribution closer to previous action samples yielding high rewards. More details on MPC and CEM are discussed in App. F.

Fig. 6(a) shows the cumulative rewards of the three methods during an episode, from which we can see that PCM performs similarly to the true dynamics and significantly outperforms PAM. To further explore why our approach works better, we calculate regret values of the evaluation of action sequences for PCM and PAM respectively. We track several planning processes and compute regret $\sum_{i=t}^{t+T} \mathbb{E}_{T^*}[r(s_i, a_i^*)] - \sum_{i=t}^{t+T} \mathbb{E}_{T^*}[r(s_i, \hat{a}_i)]$ for both PAM and PCM, where $\hat{a}_{t:t+T}$ and $a_{t:t+T}^*$ are the optimal action sequences selected by the learned model and true dynamics respectively. Regret is the difference between the real value of the action sequence selected by the model and the value of the optimal action sequence. Results in Fig. 6(b) shows that PCM has lower regret than PAM, meaning that our approach tends to pick out actions that are closer to the optimal policy.

### 5.4 ANALYSIS OF LEARNED POLICY REPRESENTATION

In this section, we conduct a study to verify whether the PCM can learn reasonable policy representations. We select several policies with different performance and feed the trajectories generated by these policies into the policy encoder module of PCM. We visualize the outputted policy representations via the t-SNE (van der Maaten & Hinton, 2008) technique in Fig. 7. We find that the policies with similar performance have similar policy representations since there is a degree of resemblance between their performed actions, while the representations of policies with widely different performance are far apart due to their quite different behavior. This result demonstrates that PCM can effectively identify similar policies and distinguish different policies. We provide results on more tasks in Appx. D.

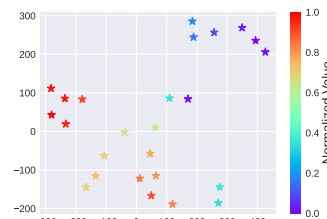

Figure 7: Visualization for policy representations of different policies learned by PCM in HalfCheetah. Points are colored according to the normalized value.

## 6 DISCUSSION AND FUTURE WORK

This paper handles the challenge that a learned dynamics model tends to have a large value gap when used to evaluate a target policy different from the data-collection policies when the offline dataset is collected by diverse behavior policies. We propose training a Policy-Conditioned Model (PCM) that generates distinct dynamics models based on different target policies. We demonstrate that PCM can achieve smaller value gaps by reducing training errors and better generalization to out-of-distribution data. Empirical results across domains and algorithms validate the superiority of our approach.

It should be noted that several possible ways exist to implement the policy-conditioned mechanism, and the RNN-based policy encoding employed in this work is just one of them. Another limitation is that we analyze the generalization of PAM and PCM based on the infinite sample assumption for each behavior policy. However, for realistic situations where only finite samples are available, the data from each policy are limited, and additional estimation errors occur in the model learning, which requires further analysis to compare PAM with PCM. In the future, we aim to find a more efficient policy representation scheme to enhance the model's generalization ability.

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

# Appendix

## Table of Contents

# A PROOFS

## A.1 VALUE GAP BOUNDS

First, we prove the bound 2 of value gap:

$$|V_{T^*}^\pi - V_{\hat{T}}^\pi| \le \frac{2R_{\max}\gamma}{(1-\gamma)^2}\mathbb{E}_{s,a\sim\rho^\pi}D_{\mathrm{TV}}(T^*(\cdot|s,a),\hat{T}(\cdot|s,a)). \tag{8}$$

*Proof.* In the proofs in this subsection, we use $\rho_{\hat{T}}^\pi$ to denote the occupancy measure under the transition model $\hat{T}$, and still use $\rho^\pi$ to denote the occupancy measure in the real dynamics.

$$|V_{T^*}^\pi - V_{\hat{T}}^\pi| = \frac{1}{1-\gamma}|\sum_{s,a}(\rho^\pi(s,a) - \rho_{\hat{T}}^\pi(s,a))r(s,a)| \tag{9}$$

$$\overset{(a)}{\le} \frac{2R_{\max}}{1-\gamma}D_{\mathrm{TV}}(\rho^\pi,\rho_{\hat{T}}^\pi) \tag{10}$$

$$\overset{(b)}{\le} \frac{2R_{\max}\gamma}{(1-\gamma)^2}\mathbb{E}_{s,a\sim\rho^\pi}D_{\mathrm{TV}}(T^*(\cdot|s,a),\hat{T}(\cdot|s,a)), \tag{11}$$

where $(a)$ holds because

$$|\sum_{s,a}(\rho^\pi(s,a) - \rho_{\hat{T}}^\pi(s,a))r(s,a)| \le \sum_{s,a}|\rho^\pi(s,a) - \rho_{\hat{T}}^\pi(s,a)||r(s,a)| \tag{12}$$

$$\le R_{\max}\sum_{s,a}|\rho^\pi(s,a) - \rho_{\hat{T}}^\pi(s,a)| \tag{13}$$

$$= 2R_{\max}D_{\mathrm{TV}}(\rho^\pi,\rho_{\hat{T}}^\pi), \tag{14}$$

and $(b)$ holds because

$$D_{\mathrm{TV}}(\rho^\pi,\rho_{\hat{T}}^\pi) \le \frac{\gamma}{1-\gamma}\mathbb{E}_{s,a\sim\rho^\pi}D_{\mathrm{TV}}(T^*(\cdot|s,a),\hat{T}(\cdot|s,a)), \tag{15}$$

which follows the proof of Lemma 11 in Xu et al. (2021). □

For comparison, we apply the traditional bound Janner et al. (2019); Xu et al. (2020) to our mixture training data distribution setting:

$$|V_{T^*}^\pi - V_{\hat{T}}^\pi| \le \frac{2R_{\max}\gamma}{(1-\gamma)^2}\mathbb{E}_{s,a\sim\rho^{\mathrm{mix}}}D_{TV}(T^*(\cdot|s,a),\hat{T}(\cdot|s,a)) \tag{16}$$

$$+ \frac{4R_{\max}}{(1-\gamma)^2}\sum_{i=1}^n w_i\max_s D_{TV}(\pi(\cdot|s),\mu_i(\cdot|s)), \tag{17}$$

which consists of the model error under the training data distribution and additional unavoidable policy divergence terms. *This result only suggests minimizing the training model error but ignores the generalization ability to the target policy.*

*Proof.*

$$|V_{T^*}^\pi - V_{\hat{T}}^\pi| \le |V_{T^*}^\pi - \sum_i w_i V_{T^*}^{\mu_i}| + |\sum_i w_i V_{T^*}^{\mu_i} - \sum_i w_i V_{\hat{T}}^{\mu_i}| + |\sum_i w_i V_{\hat{T}}^{\mu_i} - V_{\hat{T}}^\pi|$$

$$\le \sum_i w_i(|V_{T^*}^\pi - V_{T^*}^{\mu_i}| + |V_{\hat{T}}^\pi - V_{\hat{T}}^{\mu_i}|) + |\sum_{s,a}(\rho^{\mathrm{mix}}(s,a) - \rho_{\hat{T}}^{\mathrm{mix}}(s,a))r(s,a)|$$

$$\overset{(*)}{\le} \frac{2R_{\max}\gamma}{(1-\gamma)^2}\mathbb{E}_{s,a\sim\rho^{\mathrm{mix}}}D_{TV}(T^*(\cdot|s,a),\hat{T}(\cdot|s,a))$$

$$+ \frac{4R_{\max}}{(1-\gamma)^2}\sum_i w_i\max_s D_{TV}(\pi(\cdot|s),\mu_i(\cdot|s)),$$

where $(*)$ holds because

$$|V_{T^*}^\pi - V_{T^*}^{\mu_i}| \leq \frac{2R_{\max}}{(1-\gamma)^2} \max_s D_{TV}(\pi(\cdot|s), \mu_i(\cdot|s)), \tag{18}$$

$$|V_{\hat{T}}^\pi - V_{\hat{T}}^{\mu_i}| \leq \frac{2R_{\max}}{(1-\gamma)^2} \max_s D_{TV}(\pi(\cdot|s), \mu_i(\cdot|s)), \tag{19}$$

and

$$|\sum_{s,a}(\rho^{\mathrm{mix}}(s,a) - \rho_{\hat{T}}^{\mathrm{mix}}(s,a))r(s,a)| \leq \frac{2R_{\max}}{1-\gamma} D_{\mathrm{TV}}(\rho^{\mathrm{mix}}, \rho_{\hat{T}}^{\mathrm{mix}}) \tag{20}$$

$$\leq \frac{2R_{\max}}{1-\gamma} \sum_i w_i D_{\mathrm{TV}}(\rho^{\mu_i}, \rho_{\hat{T}}^{\mu_i}) \tag{21}$$

$$\leq \frac{2R_{\max}\gamma}{(1-\gamma)^2} \sum_i w_i \mathbb{E}_{s,a\sim\rho^{\mu_i}} D_{\mathrm{TV}}(T^*(\cdot|s,a), \hat{T}(\cdot|s,a)) \tag{22}$$

$$\leq \frac{2R_{\max}\gamma}{(1-\gamma)^2} \mathbb{E}_{s,a\sim\rho^{\mathrm{mix}}} D_{\mathrm{TV}}(T^*(\cdot|s,a), \hat{T}(\cdot|s,a)). \tag{23}$$

$\square$

## A.2 PROOFS IN SECTION 4

**Assumption A.1.** For the learned model $T$, the point-wise model error $D_{\mathrm{TV}}(T^*(\cdot|s,a), T(\cdot|s,a))$ is $L$-Lipschitz with respect to the state-action pairs, i.e.,

$$\left| D_{\mathrm{TV}}(T^*, T)(s_1, a_1) - D_{\mathrm{TV}}(T^*, T)(s_2, a_2) \right| \leq L \cdot D\big((s_1, a_1), (s_2, a_2)\big), \tag{24}$$

where $D(\cdot, \cdot)$ is some kind of distance defined on the state-action space $\mathcal{S} \times \mathcal{A}$.

**Proposition A.2.** *Under Assump. A.1, for any policy $\pi \in \Pi$, the model error of PAM $T_{\hat{\psi}}$ can be bounded:*

$$l(\pi, T^*, T_{\hat{\psi}}) \leq \min_{\mu_i \in \Omega} \left\{ l(\mu_i, T^*, T_{\hat{\psi}}) + L \cdot W_1(\rho^\pi, \rho^{\mu_i}) \right\}, \tag{25}$$

*where $W_1(\cdot, \cdot)$ is the Wasserstein-1 distance defined on the state-action distribution space $\mathcal{P}(\mathcal{S} \times \mathcal{A})$ with the underlying metric $D$.*

*Proof.* The Wasserstein-1 metric has a dual representation:

$$W_1(p, q) = \sup_{\|f\|_{Lip} \leq 1} \int f(x)\mathrm{d}p(x) - f(y)\mathrm{d}q(y), \tag{26}$$

where $\|f\|_{Lip}$ is the Lipschitz constant of function $f$. Therefore we have

$$|l(\pi, T^*, T_{\hat{\psi}}) - l(\mu_i, T^*, T_{\hat{\psi}})| = |\mathbb{E}_{s,a\sim\rho^\pi} D_{\mathrm{TV}}(T^*, T_{\hat{\psi}})(s,a) - \mathbb{E}_{s,a\sim\rho^{\mu_i}} D_{\mathrm{TV}}(T^*, T_{\hat{\psi}})(s,a)|$$

$$\leq \sup_{\|f\|_{Lip} \leq L} \int f(s,a)\mathrm{d}\rho^\pi(s,a) - f(s,a)\mathrm{d}\rho^{\mu_i}(s,a) \tag{27}$$

$$= L \cdot W_1(\rho^\pi, \rho^{\mu_i}). \tag{28}$$

This holds for all $\mu_i \in \Omega$, therefore

$$l(\pi, T^*, T_{\hat{\psi}}) \leq \min_{\mu_i \in \Omega}\{l(\mu_i, T^*, T_{\hat{\psi}}) + L \cdot W_1(\rho^\pi, \rho^{\mu_i})\}. \tag{29}$$

$\square$

**Definition A.3.** For a data-collection policy $\mu_i$ and a target policy $\pi$, we can define the adaptation gain of a learned policy-conditioned model:

$$C(\pi, \mu_i) := l(\pi, T^*, T_{\hat{F}(\mu_i)}) - l(\pi, T^*, T_{\hat{F}(\pi)}). \tag{30}$$

**Proposition A.4.** *Under Assump. A.1, for any policy $\pi \in \Pi$, the model error of PCM $T_{\hat{F}(\pi)}$ is bounded:*

$$l(\pi, T^*, T_{\hat{F}(\pi)}) \leq \min_{\mu_i \in \Omega} \Big\{ \underbrace{l(\mu_i, T^*, T_{\hat{F}(\mu_i)})}_{\text{training error}} + \underbrace{L \cdot W_1(\rho^\pi, \rho^{\mu_i}) - C(\pi, \mu_i)}_{\text{generalization error}} \Big\}. \tag{31}$$

*Proof.* Similar to the proof in Prop. A.2, for some behavior policy $\mu_i$, the inaccuracy of $\mu_i$-adapted model on the test distribution $\rho^\pi_{T^*}$ can be bounded by that on the training distribution $\rho^\mu_{T^*}$ and the generalization error caused by the distribution extrapolation

$$l(\pi, T^*, T_{\hat{F}(\mu_i)}) \leq l(\mu_i, T^*, T_{\hat{F}(\mu_i)}) + L \cdot W_1(\rho^\pi, \rho^{\mu_i}). \tag{32}$$

The $\pi$-adapted model $T_{\hat{F}(\pi)}$ enjoys an adaptation gain compared to $T_{\hat{F}(\mu_i)}$, compensating the extrapolation effect

$$l(\pi, T^*, T_{\hat{F}(\pi)}) = l(\pi, T^*, T_{\hat{F}(\mu_i)}) - C(\pi, \mu_i). \tag{33}$$

Therefore the generalization error of the $\pi$-adapted model $T_{\hat{F}(\pi)}$ on the test distribution $\rho^\pi$ is reduced

$$l(\pi, T^*, T_{\hat{F}(\pi)}) \leq l(\mu_i, T^*, T_{\hat{F}(\mu_i)}) + L \cdot W_1(\rho^\pi, \rho^{\mu_i}) - C(\pi, \mu_i). \tag{34}$$

This holds for all $\mu_i \in \Omega$, therefore

$$l(\pi, T^*, T_{\hat{F}(\pi)}) \leq \min_{\mu_i \in \Omega} \{ l(\mu_i, T^*, T_{\hat{F}(\pi)}) + L \cdot W_1(\rho^\pi, \rho^{\mu_i}) - C(\pi, \mu_i) \}. \tag{35}$$

$\square$

### A.3 ANALYSIS ON THE INTERMEDIATE ADAPTATION

It is hard in general to rigorously analyze the adaptation gain $C(\pi, \mu_i)$ because of the complexity of neural networks and the optimization process. To provide more concrete intuitions, in the following, we show some special cases for PCM to explain the benefit of such adaptation effects.

*Case 1: Direct match.* If the $T_{\hat{F}(\pi)}$ simply equals to $T_{\hat{F}(\mu_i)}$ for some training policies $\mu_i$, e.g., equals to the one has the smallest occupancy divergence with respect to the target $\pi$, then the adaptation gain is exactly zero, which means no adaptation effect is enabled. Hence in this case the advantage compared to PAM solely comes from the reduced error in training policy distributions.

*Case 2: Imaginary retraining.* Another example corresponds to the (probably unachievable) perfect generalization case, where $\hat{F}(\pi) = \arg\min_{\psi \in \Psi} l(\pi, T^*, T_\psi)$. In this case, we can imaginary $\rho^\pi$ based on $\pi$ and search the optimal model parameter directly. Therefore the adaptation gain completely cancels out the extrapolation error and the optimal model accuracy under the capacity limitation. This case can be regarded as a ceiling of model accuracy for any practical PCM algorithms.

*Case 3: Intermediate adaptation.* In general, we argue that the generalization ability of a well-trained PCM will fall between the two extreme cases, where the adaptation gain is greater than zero and able to partially reduce the extrapolation error within a certain region as $\pi$ diverges from $\Omega$ gradually. When $\pi$ is far enough from $\Omega$, $C$ will reach the maximum and then may start to decrease and will be less than zero finally.

We argue that the generalization ability of PCM will more resemble Case 3, in which $C$ will first increase and gradually decrease after reaching the maximum. We provide some empirical evidence in Sec. 5.2, which can support our intuition.

We provide a formulation of the intermediate adaptation effect in the following.

**Assumption A.5.** Assume that the adaptation gain $C(\pi, \mu_i)$ of a well-trained policy-conditioned model satisfies the following properties:

1. if $\mu_i = \pi$, the adaptation gain $C(\pi, \mu_i)$ equals zero, since the adaptation effect is not activated;

2. as $\pi$ diverges from $\mu_i$ gradually, the adaptation effect becomes significant, an therefore $C(\pi, \mu_i)$ increases from zero;

3. when $\pi$ is far enough from $\mu_i$, $C(\pi, \mu_i)$ reaches the maximum and then may start to decrease due to the finite samples and bounded generalization, so it leaves the effective adaptation region.

Under Assump. A.5, there exists an $L_i > 0$ such that $C(\pi, \mu_i) \geq L_i \cdot W_1(\rho^\pi, \rho^{\mu_i})$ for a data-collection policy $\mu_i$ and a target policy $\pi$ within the effective adaptation region, which can be justified by the empirical result in Fig. 3(b). Then we have

**Proposition A.6.** *Under the assumption of the adaptation gain, the generalization error of the learned policy-conditioned model can be bounded:*

$$l(\pi, T^*, T_{\hat{F}(\pi)}) \leq \min_{i \in I_\pi} \left\{ l(\mu_i, T^*, T_{\hat{F}(\mu_i)}) + (L - L_i) W_1(\rho^\pi, \rho^{\mu_i}) \right\}, \tag{36}$$

*where $I_\pi$ is the index set of the training policies in whose effective adaptation region the target policy $\pi$ lies.*

*Proof.* Substituting the assumption on the adaptation gain into the consequence of Prop. A.4 yields

$$l(\pi, T^*, T_{\hat{F}(\pi)}) \leq \min_{\mu_i \in \Omega} \left\{ l(\mu_i, T^*, T_{\hat{F}(\mu_i)}) + L \cdot W_1(\rho^\pi, \rho^{\mu_i}) - C(\pi, \mu_i) \right\} \tag{37}$$

$$\leq \min_{i \in I_\pi} \left\{ l(\mu_i, T^*, T_{\hat{F}(\mu_i)}) + (L - L_i) W_1(\rho^\pi, \rho^{\mu_i}) \right\}, \tag{38}$$

where we replace the whole data-collection policy set $\Omega$ with a local data-collection policy index set $I_\pi$ because the adaptation gain is locally assumed. $\square$

Prop. A.6 shows that the coefficient of extrapolation term is reduced from $L$ to $L - L_i$ thanks to the adaptation effect, implying better generalization to unseen policies within a reasonable effective region. Note here we do not argue that policy-conditioned models can generalize better to any target policy since here we only make local assumptions on the adaptation effect, and for those policies quite distinct from all the data-collection policies we do not expect a high fidelity no matter for policy-agnostic or policy-conditioned models.

## B    IMPLEMENTATION DETAILS

### B.1    PSEUDOCODE

The pseudocode of PCM via policy representation is listed in Alg. 1

---

**Algorithm 1** Policy-conditioned Model Learning

---

**Require:** Offline dataset $\mathcal{D} = \{\tau^{(j)}\}_{j=1}^N$ with $N$ trajectories, policy encoder $q_\phi$, policy decoder $p_\theta$, policy-conditioned dynamics model $T_\psi$.
  **for** $i = 1$ **to** $n_{\text{iter}}$ **do**
    sample a trajectory $\tau \sim \mathcal{D}$
    sample a timestep $t \sim [0, \text{len}(\tau)]$
    use the trajectory $\tau_{0:t+1}$ to optimize $\phi, \theta, \psi$ according to Eq. (5).
  **end for**

---

### B.2    IMPLEMENTATION DETAILS OF POLICY-AGNOSTIC MODEL (PAM)

PAM is a 4-layer feedforward neural network with 200 hidden units. In addition, we borrow the design of blocks in Transformer. We employ a residual connection around each of the two layers, followed by layer normalization. That is, the output of each layer is LayerNorm($x$ + Layer($x$)), where Layer(x) = Dropout(Activation(Linear($x$))).

### B.3    IMPLEMENTATION DETAILS OF POLICY-CONDITIONED MODEL (PCM)

PCM follows the same architecture as PAM except for an additional policy encoder and a policy decoder. The policy encoder is a 3-layer GRU with 128 hidden units and outputs a 128-dim embedding. Then the outputted embedding will be concatenated with the first layer's output of the dynamics model. The policy decoder takes a state and an embedding as input and outputs an action distribution. It is also constructed by a 4-layer MLP with 200 hidden units.

We present hyperparameters for model training in Tab. 3, which are shared by PAM and PCM (except $\lambda$ since it is a hyperparameter unique to PCM).

Table 3: Training hyperparameters of PAM and PCM.

| Hyperparameters | Value | Description |
| --- | --- | --- |
| Batch size | 32 | Batch size for gradient descent. |
| Optimizer | Adam | Optimizer. |
| Learning rate | 1e-4 | Learning rate for gradient descent. |
| Dropout rate | 0.1 | Dropout rate. |
| $\lambda$ | 0.01 | Weight of policy representation loss. |

## C    DETAILS OF OPE

### C.1    OFF-POLICY EVALUATION WITH PCM

Since we argue that our proposed PCM tends to have a smaller value gap, an obvious application that can make full use of its superiority is off-policy evaluation (OPE). OPE via a learned dynamics model is straightforward, which only needs to compute the return using simulated trajectories generated by the evaluated policy under the learned dynamics model. Due to the stochasticity in the model and the policy, we estimate the return for a policy with Monte-Carlo sampling. See Alg. 2 for pseudocode.

In practical evaluation, we choose $\gamma = 0.995$ and $N = 10$.

---

**Algorithm 2** Off-policy Evaluation with PCM

---

**Require:** Policy-conditioned dynamics model $(q_\phi, p_\theta, T_\psi)$ learned on $\mathcal{D}$, evaluated policy $\pi$, number of rollouts $N$. set of initial states $\mathcal{S}_0$, discount factor $\gamma$, horizon length $H$.
**for** $i = 1$ **to** $N$ **do**
 $R_i = 0$
 Sample initial state $s_0 \sim \mathcal{S}_0$
 Initialize $\tau_{-1} = \mathbf{0}$
 **for** $t = 0$ **to** $H - 1$ **do**
  $z_t = q_\phi(\tau_{t-1})$
  $a_t \sim \pi(\cdot|s_t)$
  $s_{t+1}, r_t \sim T_\psi(\cdot|s_t, a_t, z_t)$
  $R_i = R_i + \gamma^t r_t$
  $\tau_t = (\tau_{t-1}, s_t, a_t)$
 **end for**
**end for**
**return** $\frac{1}{N} \sum_{i=1}^N R_i$

---

### C.2 METRICS

The metrics we use in our paper are defined as follows:

**Absolute Error** The absolute error is defined as the difference between the value and estimated value of a policy:

$$\text{AbsErr} = |V^\pi - \hat{V}^\pi|, \tag{39}$$

where $V^\pi$ is the true value of the policy and $\hat{V}^\pi$ is the estimated value of the policy.

**Rank correlation** Rank correlation measures the correlation between the ordinal rankings of the value estimates and the true values, which can be written as:

$$\text{RankCorr} = \frac{\text{Cov}(V_{1:N}^\pi, \hat{V}_{1:N}^\pi)}{\sigma(V_{1:N}^\pi)\sigma(\hat{V}_{1:N}^\pi)}, \tag{40}$$

where $1 : N$ denotes the indices of the evaluated policies.

**Regret@k** Regret@k is the difference between the value of the best policy in the entire set, and the value of the best policy in the top-k set (where the top-k set is chosen by estimated values). It can be defined as:

$$\text{Regret @k} = \max_{i \in 1:N} V_i^\pi - \max_{j \in \text{topk}(1:N)} V_j^\pi, \tag{41}$$

where $\text{topk}(1 : N)$ denotes the indices of the top K policies as measured by estimated values $\hat{V}^\pi$.

### C.3 DETAILED RESULTS

Detailed results tables are presented here (averaged over 3 random seeds).

Table 4: Raw absolute error for each algorithm on D4RL and RL Unplugged tasks.

| Task Name | FQE | DR | IS | DICE | VPM | PAM | PCM (Ours) |
|---|---|---|---|---|---|---|---|
| halfcheetah-medium-replay | 1003±132 | 1001±129 | 1409±154 | 1440±158 | 1384±148 | 983±181 | **622±160** |
| hopper-medium-replay | 234±71 | 267±60 | 375±54 | 364±49 | 392±44 | 76±22 | **44±11** |
| walker2d-medium-replay | 313±73 | 296±54 | 427±60 | 347±51 | 424±64 | 212±39 | **140±28** |
| ant-medium-replay | **410±79** | 421±72 | 603±101 | 583±110 | 612±105 | 558±11 | 529±23 |
| halfcheetah-medium-expert | 1014±101 | 1015±103 | 1400±146 | 1078±132 | 1427±111 | 1184±421 | **935±41** |
| hopper-medium-expert | 282±76 | 426±99 | 106±29 | 259±54 | 442±43 | 180±49 | **114±5** |
| walker2d-medium-expert | 233±42 | 217±46 | 436±62 | 322±60 | 425±611 | 134±22 | **95±34** |
| ant-medium-expert | **319±67** | 326±66 | 604±102 | 471±100 | 604±106 | 524±11 | 564±81 |
| cartpole-swingup | 19±1 | 24±3 | 69±2 | 23±2 | 38±4 | 22.4±7.8 | **10±1** |
| cheetah-run | 48±2 | 40±2 | 44±2 | 23±11 | 62±4 | 11±4 | **8±1** |
| fish-swim | 20±2 | 20±2 | 35±2 | 59±2 | 31±1 | 13±0 | **11±0** |

Table 5: Normalized absolute error for each algorithm on D4RL and RL Unplugged tasks.

| Task Name | FQE | DR | IS | DICE | VPM | PAM | PCM (Ours) |
|---|---|---|---|---|---|---|---|
| halfcheetah-medium-replay | 0.50±0.06 | 0.50±0.06 | 0.75±0.08 | 0.72±0.08 | 0.69±0.07 | 0.49±0.09 | **0.31±0.08** |
| hopper-medium-replay | 0.43±0.13 | 0.49±0.11 | 0.69±0.10 | 0.67±0.07 | 0.72±0.08 | 0.14±0.04 | **0.08±0.02** |
| walker2d-medium-replay | 0.56±0.13 | 0.53±0.10 | 0.76±0.11 | 0.67±0.09 | 0.76±0.11 | 0.38±0.07 | **0.25±0.05** |
| ant-medium-replay | **0.36±0.07** | 0.37±0.06 | 0.53±0.09 | 0.51±0.10 | 0.54±0.09 | 0.49±0.01 | 0.46±0.02 |
| halfcheetah-medium-expert | 0.51±0.05 | 0.51±0.05 | 0.70±0.07 | 0.54±0.07 | 0.71±0.06 | 0.59±0.21 | **0.46±0.02** |
| hopper-medium-expert | 0.43±0.12 | 0.65±0.15 | 0.16±0.04 | 0.39±0.08 | 0.67±0.07 | 0.33±0.09 | **0.21±0.01** |
| walker2d-medium-expert | 0.42±0.08 | 0.39±0.08 | 0.78±0.11 | 0.58±0.11 | 0.76±0.11 | 0.24±0.04 | **0.17±0.06** |
| ant-medium-expert | **0.28±0.06** | 0.29±0.06 | 0.53±0.09 | 0.41±0.09 | 0.53±0.09 | 0.46±0.01 | 0.49±0.07 |
| cartpole-swingup | 0.17±0.01 | 0.22±0.02 | 0.57±0.02 | 0.19±0.01 | 0.31±0.03 | 0.20±0.07 | **0.09±0.01** |
| cheetah-run | 0.68±0.03 | 0.57±0.03 | 0.63±0.03 | 0.33±0.05 | 0.88±0.06 | 0.22±0.07 | **0.15±0.01** |
| fish-swim | 0.44±0.03 | 0.45±0.04 | 0.77±0.04 | 1.32±0.05 | 0.69±0.02 | 0.28±0.01 | **0.25±0.01** |

Table 6: Rank correlation for each algorithm on D4RL and RL Unplugged tasks.

| Task Name | FQE | DR | IS | DICE | VPM | PAM | PCM (Ours) |
|---|---|---|---|---|---|---|---|
| halfcheetah-medium-replay | 0.26±0.37 | 0.32±0.37 | 0.59±0.26 | -0.15±0.41 | -0.07±0.36 | 0.71±0.13 | **0.86±0.06** |
| hopper-medium-replay | 0.17±0.15 | 0.24±0.25 | 0.41±0.32 | 0.21±0.34 | -0.11±0.22 | 0.91±0.03 | **0.94±0.02** |
| walker2d-medium-replay | -0.19±0.36 | -0.37±0.39 | 0.65±0.24 | 0.55±0.23 | -0.52±0.25 | 0.58±0.06 | **0.71±0.16** |
| ant-medium-replay | 0.57±0.28 | 0.45±0.32 | 0.07±0.39 | -0.24±0.39 | -0.26±0.29 | -0.05±0.12 | 0.06±0.11 |
| halfcheetah-medium-expert | 0.62±0.27 | 0.62±0.27 | -0.06±0.37 | -0.08±0.35 | -0.47±0.29 | 0.44±0.46 | **0.84±0.02** |
| hopper-medium-expert | -0.33±0.30 | -0.41±0.27 | 0.37±0.27 | -0.08±0.32 | 0.21±0.32 | 0.59±0.20 | **0.74±0.03** |
| walker2d-medium-expert | 0.25±0.32 | 0.19±0.33 | 0.24±0.33 | -0.34±0.34 | 0.49±0.37 | 0.64±0.12 | **0.88±0.13** |
| ant-medium-expert | 0.37±0.35 | 0.35±0.35 | -0.21±0.35 | -0.33±0.40 | -0.28±0.28 | -0.35±0.49 | 0.02±0.12 |
| cartpole-swingup | 0.70±0.07 | 0.55±0.09 | -0.23±0.11 | -0.16±0.11 | 0.01±0.11 | 0.63±0.13 | **0.90±0.01** |
| cheetah-run | 0.56±0.08 | 0.56±0.08 | -0.01±0.12 | 0.07±0.11 | 0.01±0.12 | 0.66±0.06 | **0.74±0.03** |
| fish-swim | 0.10±0.12 | 0.11±0.12 | -0.17±0.11 | 0.44±0.09 | **0.56±0.08** | 0.20±0.13 | 0.45±0.03 |

Table 7: Regret for each algorithm on D4RL (Regret@1) and RL Unplugged (Regret@5) tasks.

| Task Name | FQE | DR | IS | DICE | VPM | PAM | PCM (Ours) |
|---|---|---|---|---|---|---|---|
| halfcheetah-medium-replay | 0.36±0.16 | 0.33±0.18 | 0.13±0.10 | 0.30±0.07 | 0.25±0.09 | **0.14±0.10** | 0.15±0.04 |
| hopper-medium-replay | 0.31±0.18 | 0.33±0.20 | 0.11±0.06 | 0.26±0.10 | 0.33±0.23 | 0.16±0.12 | **0.08±0.04** |
| walker2d-medium-replay | 0.24±0.20 | 0.68±0.23 | 0.02±0.05 | 0.18±0.12 | 0.46±0.31 | **0.01±0.01** | 0.05±0.01 |
| ant-medium-replay | **0.05±0.19** | 0.17±0.31 | 0.18±0.06 | 0.09±0.10 | 0.03±0.08 | 0.17±0.10 | 0.08±0.03 |
| halfcheetah-medium-expert | 0.14±0.07 | 0.14±0.07 | 0.73±0.42 | 0.38±0.37 | 0.80±0.34 | 0.36±0.45 | **0.12±0.05** |
| hopper-medium-expert | 0.41±0.20 | 0.34±0.35 | 0.06±0.03 | 0.20±0.08 | 0.13±0.10 | 0.14±0.14 | **0.08±0.04** |
| walker2d-medium-expert | 0.22±0.14 | 0.30±0.12 | 0.13±0.07 | 0.78±0.27 | 0.24±0.42 | 0.34±0.27 | **0.10±0.04** |
| ant-medium-expert | 0.36±0.14 | 0.37±0.13 | 0.46±0.18 | 0.60±0.16 | 0.32±0.24 | 0.59±0.28 | **0.06±0.04** |
| cartpole-swingup | 0.06±0.04 | 0.28±0.05 | 0.73±0.16 | 0.68±0.41 | 0.50±0.13 | 0.04±0.06 | **0.00±0.00** |
| cheetah-run | 0.17±0.05 | 0.09±0.05 | 0.40±0.21 | 0.27±0.05 | 0.37±0.04 | 0.24±0.18 | **0.00±0.00** |
| fish-swim | 0.50±0.03 | 0.61±0.12 | 0.12±0.05 | 0.35±0.24 | **0.02±0.02** | 0.18±0.14 | 0.11±0.06 |

# D VISUALIZATIONS FOR POLICY REPRESENTATIONS

We provide the visualization of policy representations in more tasks as shown in Fig. 8. The results shows that PCM can effectively identify similar policies and distinguish different policies.

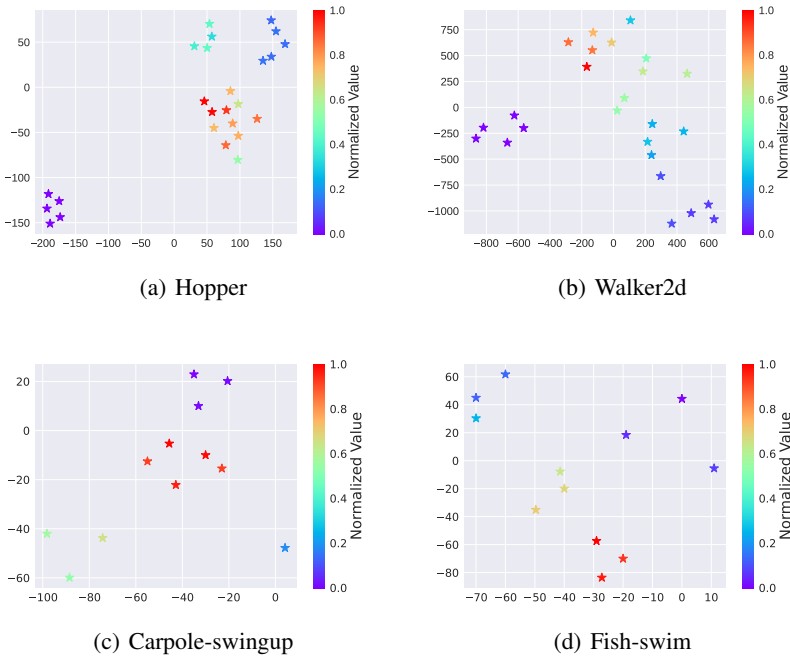

Figure 8: Illustrations of the t-SNE visualization for policy representations of different policies learned by PCM in Hopper, Walker2d, Carpole-swingup, and Fish-swim. For each task, several policies (denoted by different markers) are plotted, which are colored according to the normalized performance.

Table 8: Hyperparameters of MPC.

| Hyperparameters | Value | Description |
|---|---|---|
| Planning horizon | 30 | Length of the planning horizon. |
| Num candidates | 500 | Number of the sampled sequence actions. |
| Num elites | 50 | Number of elites. |
| $\alpha$ | 0.1 | How much of the previous mean is used for the next CEM iteration. |
| CEM iteration | 25 | Number of the CEM iterations. |

## E  EXPERIMENT DETAILS OF THE EMPIRICAL EVIDENCE VERSIFICATION

We now introduce some experimental details in Sec 5.2.

### E.1  DETAILS OF DATA COLLECTION

**Data collection for evaluating model error and adaptation gain.** We train SAC (Haarnoja et al., 2018) for 1000 epochs (each epoch contains 1k gradient steps) in HalfCheetah environment. Then we record policy snapshots at 10, 20,..., 100 epochs and use each policy to sample 20 trajectories for data collection. When evaluating adaptation gain, we fix a data-collection policy $\mu_i$ and select various policies during SAC training as target policies.

**Data collection for evaluating value gap.** We construct datasets with varying levels of diversity (20%, 50%, 80%), where the percentage indicates that the dataset is created from the replay buffer of SAC (Haarnoja et al., 2018) until the policy reaches the specific level of performance. We train PAM and PCM on each dataset and test them on the other 11 policies provided by the DOPE benchmark Fu et al. (2021a), which were unseen before in the datasets.

### E.2  DETAILS OF ADAPTATION GAIN

Recall that the adaptation gain is defined as

$$C(\pi, \mu_i) := l(\pi, T^*, T_{\hat{F}(\mu_i)}) - l(\pi, T^*, T_{\hat{F}(\pi)}),$$

where $l$ is the TV divergence between true and learned dynamics. However, directly computing $C$ is intractable since the true transition $T^*$ is unknown. Therefore, we instead use the mean squared error $\frac{1}{|\mathcal{S}|} \sum_{i=1}^{|\mathcal{S}|} \left( \hat{s}^{(i)} - s^{(i)} \right)^2$ to compute inconsistency between true and learned dynamics, where $\hat{s} \sim T_\psi$ and $s \sim T^*$.

## F  DETAILS OF MODEL PREDICTIVE CONTROL

Once a model is learned, we can use it for control by predicting the future outcomes of candidate policies or actions and then selecting the particular candidate that is predicted to result in the highest reward. A classical planning method is model predictive control (MPC) (Camacho & Alba, 2013) which plans for a sequence of actions. We choose MPC here for several reasons, including implementation simplicity and lower computational burden.

Given the state of the system $s_t$ at time $t$, the planning horizon $H$, and an action sequence $\boldsymbol{a}_{t:t+H} = \{a_t, ..., a_{t+H}\}$, the dynamics model $T_\psi$ will predict a state trajectory $\boldsymbol{s}_{t:t+H}$. At each time step $t$, the MPC controller applies the first action $a_t$ of the sequence of the optimized actions $\arg\max_{\boldsymbol{a}_{t:t+H}} \sum_{i=t}^{H} r(s_i, a_i)$. A common way to generate candidate action sequences is random shooting, which simply generates $N$ independent random action sequences. This approach has been shown to achieve success on many control tasks, but it has significant drawbacks: it scales poorly with the dimension of both the action space and the planning horizon. As suggested in Chua et al. (2018), we use CEM instead of random shooting, which samples actions from a distribution closer to previous action samples that yielded high rewards.

All the experiments in Sec. 5.3.3 share the same hyperparameters as shown in Tab. 8.

## G  DETAILS OF OFFLINE POLICY SELECTION (OPS)

We train MOPO (Yu et al., 2020) for 1000 epochs and record policy snapshots at the latest 20 epochs for OPS. We compare our method against FQE, IS, DICE, PAM as well as directly selecting the last-epoch policy. The raw performance of policies selected by each OPS approach on each task is listed in Tab. 9. Tab. 2 shows the performance gains by different methods. The performance gain is computed by $\frac{(V_{\text{selected}} - \bar{V})}{V_{\text{max}} - \bar{V}} \times 100\%$, where $V_{\text{selected}}$ represents the value of the selected policy and $\bar{V}, V_{\text{max}}$ are the average and max values of the evaluated policies, respectively. It is noteworthy that the gains of FQE and PAM are even lower than directly selecting the last-epoch policy, which is also indicated in another work (Qin et al., 2022). In contrast, our approach shows a brilliant performance, implying that it can reliably choose a better policy for an offline RL algorithm to deploy.

Table 9: Raw performance of offline policy selection for MOPO (Yu et al., 2020) via different approaches.

| Task Name | Last Epoch | FQE | IS | DICE | PAM | PCM (Ours) |
|---|---|---|---|---|---|---|
| halfcheetah-medium-replay | 72.3 | 70.0 | 74.8 | 71.3 | 71.3 | 75.9 |
| hopper-medium-replay | 102.0 | 100.0 | 102.0 | 98.7 | 101.2 | 102.8 |
| walker2d-medium-replay | 79.3 | 80.3 | 83.5 | 81.2 | 76.5 | 85.4 |
| Average | 84.5 | 83.4 | 86.8 | 83.7 | 83.0 | 88.0 |

## H  SOCIETAL IMPACT

This work studies a method for environment model learning. Reconstructing an accurate environment of the real world will promote the wide adoption of decision-making policy optimization methods in real life, enhancing our daily experience. We are aware that decision-making policy in some domains like recommendation systems that interact with customers may have risks of causing price discrimination and misleading customers if inappropriately used. A promising way to reduce the risk is to introduce fairness into policy optimization and rules to constrain the actions. We are involved in and advocating research in such directions. We believe that business organizations would like to embrace fair systems that can ultimately bring long-term financial benefits by providing a better user experience.

## APPENDIX REFERENCES

Justin Fu, Mohammad Norouzi, Ofir Nachum, George Tucker, Ziyu Wang, Alexander Novikov, Mengjiao Yang, Michael R. Zhang, Yutian Chen, Aviral Kumar, Cosmin Paduraru, Sergey Levine, and Thomas Paine. Benchmarks for deep off-policy evaluation. In *9th International Conference on Learning Representations (ICLR'21)*, virtual event, 2021.

Michael Janner, Justin Fu, Marvin Zhang, and Sergey Levine. When to trust your model: Model-based policy optimization. In *Advances in neural information processing systems 32 (NeurIPS'19)*, Vancouver, BC, Canada, 2019.

Tian Xu, Ziniu Li, and Yang Yu. Error bounds of imitating policies and environments. In *Advances in Neural Information Processing Systems 33 (NeurIPS'20)*, virtual event, 2020.

Tian Xu, Ziniu Li, and Yang Yu. Error bounds of imitating policies and environments for reinforcement learning. *IEEE Transactions on Pattern Analysis and Machine Intelligence*, 44(10): 6968–6980, 2021.

