# OpenReview forum: "Effective Offline Environment Reconstruction when the Dataset is Collected from Diversified Behavior Policies"
_ICLR.cc/2024/Conference — Submitted to ICLR 2024_

### Official Review · Reviewer_U3YJ · 2023-10-31

**Soundness:** 2 fair
**Presentation:** 3 good
**Contribution:** 2 fair
**Rating:** 3
**Confidence:** 3

**Summary:**

This paper studies the offline policy evaluation (OPE) problem. The authors propose Policy-Conditioned Model (PCM) learning, which learns a dynamics model that depend on the policy to be evaluated. In contrast, traditional model learning learn a unified dynamics model from a dataset that is agnostic to policy. Theoretical analysis are provided to justify the effectiveness of PCM. Experiments are also conducted to show the advantage of the proposed method over prior works.

**Strengths:**

1. This paper is well-written and easy to follow.
2. The intuition is straightforward and the algorithm is neat.
3. The theoretical part provide mathematical justification for the intuition of the algorithm.

**Weaknesses:**

1. The Related Work does not seem comprehensive enough to me. It should include more recent works on OPE and MBORL. I did not find citation of OPE later than 2020. For MBORL, COMBO, ROMI and many more later works are missing. I suggest adding a more comprehensive Related Work.
2. I am not fully convinced by the experiment. As mentioned in the paper, PCM adopt more advanced network architecture to enhance performance so that even the PAM achieves better performance. Does this mean that the advantage of PCM may not come from the algorithmic novelty but from the better network architecture instead?
3. As an experiment paper, I do not think the experiments conducted are extensive enough, more tasks should be tested.

Minor Mistakes:
Figure 2 and 3 seem to have spacing issues. The legends are partially screened by the captions
In some formulae (e.g. 5 and 8), the parentheses should wrap the content inside.

**Questions:**

1. Can the authors explain Figure 2(c) in more detail, how are the quantities calculated?
2. Can the authors explain the experiment methodology in more detail, as mentioned in Weakness 2
3. In Figure 3, what does the data for percentage 0% and 100% look like. Also, the differences of value gaps does not seem substantial to me, is it the case?
4. How does PCM perform on other tasks besides mujoco? Specifically, D4RL has maze, antmaze, carla etc. besides mujoco.

**Details Of Ethics Concerns:**

The authors touched on this subject in the last section of the Appendix

---

> ### Author Response · Authors · 2023-11-20
> **Official response**
>
> Thanks for the insightful suggestions on our paper. We response the questions and concerns as follows:
>
> ### **Q1: The Related Work does not seem comprehensive enough to me. It should include more recent works later than 2020.**
>
> Thank you for the suggestion. We have included additional related work in the main text, covering topics such as COMBO, ROMI, and others later than 2020.
>
> ### **Q2: As mentioned in the paper, PCM adopt more advanced network architecture to enhance performance so that even the PAM achieves better performance. Does this mean that the advantage of PCM may not come from the algorithmic novelty but from the better network architecture instead?**
>
> Thank you for your critical observation. We understand the concern that the improvements might be attributed to advanced network architectures rather than the algorithmic novelty of PCM itself.
>
> To address this concern, we emphasize that both PCM and PAM employ identical network architectures in all our experiments, as detailed in Appendix B.2 of our paper. Specifically, both models use a 4-layer feedforward MLP with 200 hidden units per layer. Each layer follows the structure LayerNorm(x + Dropout(Activation(Linear(x)))). The primary distinction between PCM and PAM lies in the addition of a policy encoder in PCM. This encoder generates a policy embedding, which is then used as an additional input for PCM. However, this is the only architectural difference, ensuring that any performance discrepancies are attributable to this specific module rather than overall architectural enhancements.
>
> Our results in Table 1 in the mainbody substantiate this point. They demonstrate that both the shared architecture and the unique policy-conditioned module of PCM are instrumental in reducing value gaps.  This indicates that the performance advantage of PCM is indeed a result of the algorithmic innovation introduced by the policy-conditioned module, rather than merely an outcome of an advanced network architecture.
>
> Furthermore, to bolster our comparison and validate the efficacy of PCM, we have also implemented PCM with a standard MLP architecture. The results of this implementation can be found in the general response. This implementation verify that the PCM mechanism consistently aids in reducing value gaps across different architectures.
>
>
> ### **Q3: I do not think the experiments conducted are extensive enough, more tasks should be tested.**
>
> Thanks for your valuable suggestion. We have supplemented five datasets for OPE task, including ant-medium-replay, ant-medium-expert, halfcheetah-medium-expert, hopper-medium-expert, and walker2d-medium-expert. Similarly, our method also outperforms other methods significantly on these tasks.  Please see Table 3 to Table 6 for raw results in our revised paper and full discussion in the general response.
>
> ### **Q4: Can the authors explain Figure 2(c) in more detail, how are the quantities calculated?**
>
> In our experiment, we constructed a set of 10 distinct policies. These policies were then used to generate datasets on which our PCM was trained, resulting in 10 unique vector policy embeddings corresponding to each policy. We evaluated the performance of each policy using models conditioned on each of the 10 embeddings. The value gap was calculated for each pairing, providing a matrix of outcomes.
>
> Specifically, in Figure 2(c), each cell in row i, column j, quantifies the value gap when policy j is evaluated using the PCM adapted with embedding i. The heatmap's intensity indicates the magnitude of the value gap, with darker colors representing larger gaps. This visual representation allows us to observe how well the model, when conditioned with a specific policy embedding, can adapt and evaluate other policies.
>
> ### **Q5: In Figure 3, what does the data for percentage 0% and 100% look like. Also, the differences of value gaps does not seem substantial to me, is it the case?**
>
> We would like to directly compare the value gaps of PAM and PCM and also investigate the influence of different levels of dataset diversity on them. To do so, we construct datasets with varying levels of diversity (0\%, 20\%, 50\%, 80\%, 100\%), where the percentage indicates that the dataset is created from the replay buffer of SAC until the policy reaches the specific level of performance.
>
> ### **Q6: How does PCM perform on other tasks besides mujoco? Specifically, D4RL has maze, antmaze, carla etc. besides mujoco.**
>
> The reason for not including maze, antmaze, carla, etc. is that these datasets are collected by a single policy, which is inconsistent with the diversified nature of datasets we discuss in our paper.  However, we agree that more tasks should be evaluated in our experiments, and as mentioned in response to Q3, we have included more tasks for evaluation.
>
> ### **Q7: Figure 2 and 3 seem to have spacing issues.**
>
> Thanks for pointing out this formatting issue. We have fixed them in our revised paper.

---

> > ### Comment · Reviewer_U3YJ · 2023-11-21
> >
> > Thank you for the explanations!
> >
> > I might have missed some parts of the revised paper but I did not find you including COMBO, ROMI, etc. Could you tell me where exactly are they? As far as I know, the list of popular MBORL can be long.
> >
> > I appreciate that the authors added more experiments. I understand that given the rebuttal time constraint, it is impossible to add many experiments. But still, I think as an empirical work, the evaluation should be more comprehensive. For example, how does the algorithm perform on the 'random' datasets? I do think there are some valuable points in this paper but comprehensive evaluation is also important. It is all right if the performances are not the best if the authors provide reasonable explanations.
> >
> > I choose to retain my score and I encourage the authors conduct further investigation on this method.

---

> ### Author Response · Authors · 2023-11-21
>
> Thank you for your prompt response and continued engagement with our work. We appreciate your valuable insights and would like to address the concerns raised.
>
> 1. **Inclusion of COMBO, ROMI, etc. in Related Work**: We apologize for the oversight in our previous submission where the updated related work section was mistakenly omitted. We have rectified this error and submitted a new version of the manuscript. The updated related work now includes a comprehensive discussion on models such as COMBO, ROMI, and others, reflecting a broader spectrum of popular MBORL approaches.
>
> 2. **Comprehensive Evaluation Concerns**: We acknowledge your concern regarding the comprehensiveness of our evaluation. We want to clarify that the 'random' dataset does not align with the assumptions of our approach, as it is generated from a single random policy rather than the multi-source policies our method is designed for. After adding the results in medium-expert datasets (thanks to the reviewer BstF's suggestion), we believe that our experiments **have covered all tasks in the D4RL dataset that involve multi-source policy collections**. We believe these additions make our experimental evaluation more robust and comprehensive.
>
> We are committed to conducting further investigations to refine our method and extend its applicability. We hope that the revisions and clarifications provided address your concerns and demonstrate the rigor of our approach. Should you have any additional questions or require further clarification, we are readily available to engage in further discussions.

---

### Official Review · Reviewer_BstF · 2023-11-01

**Soundness:** 3 good
**Presentation:** 3 good
**Contribution:** 2 fair
**Rating:** 5
**Confidence:** 3

**Summary:**

The paper proposes a policy-conditioned dynamics model for off-policy evaluation. The proposed methods achieved good performance on the robotic tasks, with bounded model error under their assumptions.

**Strengths:**

S1.	The paper is well written, especially introduction and methodology are clearly described.

S2.	 The proposed method achieves better results compared to baselines, on selected tasks in robotic tasks.

**Weaknesses:**

W1.	The ideas of assuming mixed-type of data and leveraging evaluation policies to improve state-action visitation are not very interesting. Similar assumptions and problems were investigated by recent OPE work [1], which also conducted experiments on D4RL. The major claim, “model learning from offline datasets collected by numerous behavior policies in fact implies a problem of data fitting from a mixture of multi-source data distributions, which is ignored in current model learning paradigms (in Introduction) looks overstrong, which should be carefully scoped, given there are already works investigating such scenarios (e.g., [1-2]).

W2.	Experiments may not be thoroughly designed.

(i) Only using 3 tasks out of overall 20 tasks in D4RL environment and 3 out of 9 RLUnplugged environment, without a well reason. The main content doesn’t accurately or explicitly describe such experimental settings, given they do not strictly follow the referred benchmark DOPE. In D4RL, Ant-related tasks are totally ignored. Though authors describe that they consider data collected by diverse policies, it may not be a strong reason for not using all types of tasks to evaluate the robustness and effectiveness of the proposed method. At least “medium-expert” tasks, which are mixed-type datasets, should be included as well.

(ii) For off-policy selection, some baselines (e.g., IS) are not included. I think they are still able to be compared, by using the policy with the maximum OPE values. For off-policy evaluation, it might be also helpful to evaluate if recent works with similar ideas can be considered, as mentioned in (1).


W3.	Reproducibility is hard to evaluate. There is no code supplementary. Raw results of absolute error without normalization are not provided as in [1], raw results of each task are not provided. Those makes it hard to be further utilized by interested researchers and limit potential impacts.


Minor:
1.	Figure 3: Part of labels on x axis is covered by captions.

References
[1] Gao, Q., Gao, G., Chi, M., & Pajic, M. (2022, September). Variational Latent Branching Model for Off-Policy Evaluation. In The Eleventh International Conference on Learning Representations.

[2] Zhang, G., & Kashima, H. (2023, June). Behavior estimation from multi-source data for offline reinforcement learning. In Proceedings of the AAAI Conference on Artificial Intelligence (Vol. 37, No. 9, pp. 11201-11209).

**Questions:**

See Weaknesses

---

> ### Author Response · Authors · 2023-11-20
> **Official response**
>
> Thanks for the insightful suggestions on our paper. We response the questions and concerns as follows:
>
> ### **Q1: The ideas of assuming mixed-type of data and leveraging evaluation policies to improve state-action visitation are not very interesting. Similar assumptions and problems were investigated by recent OPE work [1], which also conducted experiments on D4RL. The major claim, “model learning from offline datasets collected by num... current model learning paradigms" (in Introduction)  looks overstrong, which should be carefully scoped, given there are already works investigating such scenarios (e.g., [1-2]).**
>
> Thank you for your insightful comments and the referenced articles. Upon careful consideration, we maintain the validity of our claim and provide the following clarifications:
>
> Regarding the work in [1], while it shares similarities with our approach in encoding historical data, the purposes differ significantly. [1] focuses on “learning a compact and disentangled latent representation space from offline trajectories, which can better capture the dynamics underlying environments”. In contrast, our work is centered on obtaining representations for policy representation. This divergence in objectives leads to distinct training methodologies: [1] employs a variational inference framework to construct an ELBO for a robust environmental representation space, whereas we use the representation as an extra input for model prediction, aiming solely at reconstructing the policy to encode its representation. As we clarified to ****reviewer Ag2q****, our contribution does not lie in the specific technical implementation of using RNNs to encode trajectory features. To our knowledge, this particular application of policy-conditioned methods to enhance policy learning in the context of diverse offline RL datasets is a novel contribution. It surpasses existing methods by focusing on aligning policies with the prediction emphasis of dynamic models, deviating from the primarily trajectory-encoding approach in prior research.
>
> Regarding [2], the discussion revolves around the diversity of dataset sources in a model-free offline policy learning setting. In this context, the issue of policy being multi-source is intuitively obvious, as the direct goal is to imitate and optimize the offline data's policy. However, in a model-based scenario, the hidden multi-source data distribution caused by multiple source policies and the potential to leverage this feature for enhanced model generalization are not as readily apparent.
>
> In summary, our assertion that model learning from offline datasets collected by numerous behavior policies implies a problem of data fitting from a mixture of multi-source data distributions remains unaddressed in current literature. Therefore, our claim is not overstated but rather highlights a novel aspect that has not been formally proposed until now.
>
>
> ### **Q2: In D4RL, Ant-related tasks are totally ignored. Though authors describe that they consider data collected by diverse policies, it may not be a strong reason for not using all types of tasks to evaluate the robustness and effectiveness of the proposed method. At least “medium-expert” tasks, which are mixed-type datasets, should be included as well.**
>
> Thanks for the valuable suggestion. We have supplemented five datasets for OPE task, including ant-medium-replay, ant-medium-expert, halfcheetah-medium-expert, hopper-medium-expert, and walker2d-medium-expert. Similarly, our method also outperforms other methods significantly on these tasks. Please see Table 3 to Table 6 for raw results in our revised paper and full discussion in the general response.
>
> ### **Q3: For off-policy selection, some baselines (e.g., IS) are not included. I think they are still able to be compared, by using the policy with the maximum OPE values. For off-policy evaluation, it might be also helpful to evaluate if recent works with similar ideas can be considered, as mentioned in (1).**
>
> Thanks for the valuable suggestion. In previous version, we selected FQE as the baseline for comparison since it the best baseline in OPE tasks. Following the suggestion, we have additionally evaluated IS and DICE for off-policy selection, which is shown in Page 23 Table 8-9 in the appendix, and Table 2 in the main body. We observe that IS indeed has more competitive performance (59.4% performance gain) than FQE (20.6% performance gain) but still falls behind PCM (71.7% performance gain).
>
> ### **Q4: Reproducibility is hard to evaluate. There is no code supplementary. Raw results of absolute error without normalization are not provided as in [1], raw results of each task are not provided.**
>
> We are glad to opensource our code after accpetion. Raw results are on Page 19-20 (Table 4-7) in our paper.
>
> ### **Q5: Figure 3: Part of labels on x axis is covered by captions.**
>
> Thanks for pointing out this formatting issue. We have fixed them in our revised paper.

---

### Official Review · Reviewer_ENii · 2023-11-01

**Soundness:** 2 fair
**Presentation:** 3 good
**Contribution:** 3 good
**Rating:** 6
**Confidence:** 4

**Summary:**

In reinforcement learning, it is crucial to have an accurate environment dynamics model to evaluate different policies' value in tasks like offline policy optimization and policy evaluation. However, the learned model is known to have large value gaps when evaluating target policies different from data-collection policies. This issue has hindered the wide adoption of models as various policies are needed for evaluation in these downstream tasks. In this paper, the authors focus on one of the typical offline environment learning scenarios where the offline datasets is collected from diversified policies. They utilize an implicit multi-source nature in this scenario and propose an easy-to-implement yet effective algorithm, policy-conditioned model (PCM) learning, for accurate model learning. PCM is a meta-dynamics model that is trained to be aware of the evaluation policies and on-the-fly adjust the model to match the evaluation policies' state-action distribution to improve the prediction accuracy. They provide a theoretical analysis and experimental evidence to demonstrate the feasibility of reducing value gaps by adapting the dynamics model under different policies. Experiment results show that PCM ourperforms the existing SOTA off-policy evaluation in the DOPE benchmark with *a large margin*, and derives significantly better policies in offline policy selection and model prediction control compared with the standard model learning method.

**Strengths:**

1. The environment model learning under the setting of dataset is collected from diversified behavior policies is an interesting topic and has positive contribution to the research community. This problem is actually common in practice considering that the large decision mdel often relies on the world model reconstructed from the diversed policy behaviors.
2. The description of problem modeling and analysis is clear. The whole paper is generally well organized.
3. The proposed solution in this paper is concise enough and it will be enlightening for subsequent research.

**Weaknesses:**

1. The universality of the dataset collected from diversified behavior policies has not been comprehensively elaborated which limits the meaning of this problem.
2. The relationship between the proposed solution and the challenges in the setting of dataset collected from diversified behavior policies is not clarified well. It is still unknown or at least confusing the difference between the setting of the dataset collected from homogeneous behavior policies and the dataset collected from diversified behavior policies. And what are the additional challenges brought by this new setting and how the proposed approach help solve these challenges?
3. Though the authors have paid much attention to how to adapt the learned environment model for various policies in different downstream tasks, how to overcome the challenges of the diverse upstream dataset is less covered in the paper.
4. The experiments in this paper is still not sufficient enough. Though conducting experiments on different tasks and providing corresponding insightful analysis, the authors are suggested to introduce more environments and baselines to help demonstrate the applicability and superiority of the proposed method.

**Questions:**

1. The authors are encouraged to provide more visualizations to the representations of different learned policies by PCM in various environments.
2. The authors are expected to provide more evidence and supporting materials on why choosing OPE, OPS, and MPC as the downstream tasks to validate the effectiveness of the proposed method.
See the weaknesses above.

---

> ### Author Response · Authors · 2023-11-20
> **official response (part 1)**
>
> Thank you for all of the insightful questions on our paper. We response the questions and concerns as follows:
>
>
> ### **Q1: The universality of the dataset collected from diversified behavior policies has not been comprehensively elaborated which limits the meaning of this problem.**
>
> Thank you for your valuable feedback regarding the universality of the dataset derived from diversified behavior policies in our research. We acknowledge the importance of this aspect and have addressed it more comprehensively in our revised manuscript.
>
> In practical scenarios, the multi-source nature of offline datasets is a common phenomenon, particularly in fields such as robotic manipulation, autonomous driving, and sequential recommendation. These domains often involve data collected from a diverse array of policies. These include parameterized policies, rule-based policies, and human behavior policies, each contributing unique and realistic dimensions to the dataset.
>
> In our revised paper, we have expanded the introduction to include a detailed discussion of this aspect, illustrating the relevance and applicability of our approach to real-world scenarios. To support our argument, we have referenced several key studies:
>
> 1. "Roboturk: A crowdsourcing platform for robotic skill learning through imitation" and "What matters in learning from offline human demonstrations for robot manipulation" [1,2] highlight the significance of varied data sources in robotic manipulation.
> 2. "BDD100K: A diverse driving video database with scalable annotation tooling" and "Scalability in perception for autonomous driving: Waymo open dataset" [3,4] underscore the diversity in autonomous driving datasets.
> 3. "Open bandit dataset and pipeline: Towards realistic and reproducible off-policy evaluation" and "Kuairec: A fully-observed dataset and insights for evaluating recommendation systems" [5,6] elucidate the variety in datasets used for sequential recommendation.
>
> These references substantiate the universality and practical importance of multi-source datasets in offline RL research, aligning closely with the objectives and scope of our study.
>
> We believe that this enhanced discussion in our paper now adequately addresses the concern raised and underscores the significance and applicability of our research in diverse real-world contexts.
>
> [1] Roboturk: A crowdsourcing platform for robotic skill learning through imitation.
>
> [2] What matters in learning from offline human demonstrations for robot manipulation.
>
> [3] BDD100K: A diverse driving video database with scalable annotation tooling.
>
> [4] Scalability in perception for autonomous driving: Waymo open dataset.
>
> [5] Open bandit dataset and pipeline: Towards realistic and reproducible off-policy evaluation.
>
> [6] Kuairec: A fully-observed dataset and insights for evaluating recommendation systems.
>
> ### **Q2: It is still unknown or at least confusing the difference between the setting of the dataset collected from homogeneous behavior policies and the dataset collected from diversified behavior policies. And what are the additional challenges brought by this new setting and how the proposed approach help solve these challenges?**
>
> Thank you for the valuable comments. In our revised manuscript, we have expanded our discussion in this regard, particularly in the methodology section, to address these concerns more comprehensively.
>
> The key distinction in our research lies in the comparison between datasets collected from homogeneous behavior policies and those from diversified behavior policies. Traditional approaches often rely on the simplification of using data from a single-behavior policy, which does not fully capture the complexity and variability found in real-world applications. In contrast, our study focuses on the multi-source nature of data, which is more representative of realistic scenarios, as explained in our previous response.
>
> **The unique aspect of our approach is not about addressing new challenges presented by diversified datasets but rather about leveraging the additional information inherent in such settings.** In a multi-source dataset, different trajectories may originate from varying policies, but the transition data within each trajectory are consistently generated by the same policy. This distinct feature provides valuable insights for dynamics model learning, which is a central aspect of our proposed method.
>
> By acknowledging and utilizing this additional information, our approach goes beyond the limitations of  algorithms designed for single-source datasets. It recognizes and exploits the nuances in multi-source data that are often overlooked in more simplified formulations. We believe this refined perspective, now thoroughly discussed in our paper, underscores the novelty and relevance of our research in addressing the complexities of real-world data in AI applications.

---

> ### Author Response · Authors · 2023-11-20
> **official response (part 2)**
>
> ### **Q3: Though the authors have paid much attention to how to adapt the learned environment model for various policies in different downstream tasks, how to overcome the challenges of the diverse upstream dataset is less covered in the paper.**
>
> Thank you for your valuable observation regarding our paper's focus on adapting learned environmental models to various policies in different downstream tasks.
>
> We would like to clarify that our research fundamentally revolves around leveraging the diversity of datasets to learn a more accurate and faithful environmental model. The diverse nature of the dataset is not just a condition for our method but rather the cornerstone upon which our approach is built. Hence, our primary aim is to demonstrate how to effectively utilize the inherent diversity in datasets to enhance environmental model learning.
>
> Given this focus, the challenges posed by the diverse nature of the dataset are integral to our research. However, the explicit discussion on overcoming these challenges may appear less prominent as our method inherently embraces and capitalizes on this diversity, rather than viewing it as a hurdle to be overcome.
>
> We appreciate your input and have made efforts in our revised manuscript to articulate this aspect more clearly, ensuring that our approach's alignment with the premise of diverse datasets is well understood.
>
> ### **Q4: Though conducting experiments on different tasks and providing corresponding insightful analysis, the authors are suggested to introduce more environments and baselines to help demonstrate the applicability and superiority of the proposed method.**
>
> Thank you for your constructive suggestion regarding the scope of experiments in our paper. In response to your feedback, we have expanded our experimental framework by including additional datasets that further illustrate the applicability and effectiveness of our proposed method.
>
> In our revised manuscript, we have incorporated results from five new datasets for the Off-Policy Evaluation (OPE) task. These datasets are ant-medium-replay, ant-medium-expert, halfcheetah-medium-expert, hopper-medium-expert, and walker2d-medium-expert. The inclusion of these diverse environments enables a more comprehensive evaluation of our method across different contexts and challenges.
>
> Our updated results, which can be found in the general response, demonstrate that our method continues to significantly outperform other methods on these additional tasks. This further substantiates our method's superiority and versatility in various settings.
>
> We believe that these enhancements to our experimental section address your concerns and effectively showcase the robustness and generalizability of our approach.
>
> ### **Q5: The authors are encouraged to provide more visualizations to the representations of different learned policies by PCM in various environments.**
>
> These new visuals provide a clearer and more comprehensive representation of how different learned policies by PCM perform in diverse environments. You can find these enhancements on Page 21, where we have meticulously illustrated the dynamics of different policies in action. The results show that PCM can effectively identify similar policies and distinguish different policies.  We believe that these visual aids will significantly augment the reader's understanding and appreciation of our method's capabilities.
>
> ### **Q6: The authors are expected to provide more evidence and supporting materials on why choosing OPE, OPS, and MPC as the downstream tasks to validate the effectiveness of the proposed method. See the weaknesses above.**
>
> These tasks correspond to the discussion in related works. OPE is the task most relevant to the “value gap”. Since OPE needs to evaluate the value of a target policy and therefore we can expect that a more accurate model that has a smaller value gap should do better in OPE. OPS is a task derived from offline RL. For the offline RL setting, it tends to be not allowed to select a desirable policy from many candidates via evaluating them in the real environment, thus OPS is a key process for offline RL, which is also be attain great importance in other work [7]. MPC is a task to consider whether our model can be applied to decision/control rather than only policy evaluation. We don’t consider policy learning in our paper since policy learning with a learned model in the offline setting always needs to incorporate some additional designs, like ensemble, uncertainty quantification, and penalty  [8, 9], which introduce more variables and make it difficult to evaluate whether the improvement of policy learning is due to the more accurate model.
>
> [7] NeoRL: A near real-world benchmark for offline reinforcement learning.
>
> [8] MOPO: model-based offline policy optimization.
>
> [9] Model-Bellman inconsistency for model-based offline reinforcement learning.

---

### Official Review · Reviewer_Ag2q · 2023-11-01

**Soundness:** 2 fair
**Presentation:** 3 good
**Contribution:** 2 fair
**Rating:** 3
**Confidence:** 4

**Summary:**

The paper proposes a method called PCM to tackle the complex model learning process in RL for offline policy optimization and evaluation. When offline dataset is collected by a diverse set of behavior policies, it might be challenging to train a single dynamics model with tolerable error due to model capacity. To this end, PCM proposes to divide the dataset into subtasks associated with different behavior policies, and train policy-conditioned dyanamics model to improve the value gap.

**Strengths:**

1. The paper proposes to reduce modeling error of the MDP dynamics by learning policy-conditioned models associated with different data-collecting behavior policies, which is connected to popular context-based meta-learning paradigms such as [1][2][3].

2. The motivating example of "Varied dynamics models for different policies" in section 4.3 is interesting.

3. The authors conduct a wide range of experiments to demonstrate the superiority of proposed PCM.

[1] Efficient off-policy meta-reinforcement learning via probabilistic context variables.

[2] Focal: Efficient fully-offline meta-reinforcement learning via distance metric learning and behavior regularization.

[3] Fast Adaptation to New Environments via Policy-Dynamics Value Functions.

**Weaknesses:**

1. **Limited technical contributions**.  The central idea of the paper is training policy-conditioned dynamics models instead of a single model on a diverse offline RL dataset to achieve better modeling error. However, the idea of using MDP-conditioned embeddings for training a universal model in deep RL is not new, see for example [9][10][11].

2. **Gap between theory and experiments**: I also find the main theorem of the paper, Proposition 4.2 a relatively trivial result. On the RHS of (8), the training error minus the adaptation gain is precisely the model error on the LHS. It only shows that the generalization error can be bounded given a Lipshitz assumption, without explictly modeling the effect of policy embedding. The main argument seems to be that the adaptation gain of PCM is positive but zero for PAM. However, one can simply fine-tune PAM given the data of a new policy $\pi$ to achieve a positive adaptation gain, which is also a stronger baseline should be compared in experiments in 5.3.1 (see point 3). Therefore, **it's not convincing enough why policy-conditioned PCM is superior given the theory**, which is supposed to be the main contribution of the paper.

3. **Missing important model-based baselines**. As mentioned in 5.3.1, naive PAM significantly outperforms other competitive baselines of off-policy evaluation because the authors use better architectures than MLP. Thanks for being honest and transparent about this point. However, since the key contribution PCM is policy-conditioned design of dynamics model, I believe it's important to see whether PCM outperforms other competitive **model-based OPE baselines when using the same architecture**. A simple finetuned PAM on unseen task should also be compared as a stronger basline (see point 2).

4. The review of related work of model-based offline RL is inadequate. In this submission, the related work only refers to those published before 2021 and there are more recently published works such as PMDB[1], VMG[2], RAMBO[3], COUNT-MORL[4], ROSMO[5], CABI[6], AMPL[7], CBOP[8]. I personally do not agree with the authors' perspective that MBORL are generally categorized into two groups: MPC and PL. For example, AMPL[7] focuses on the model learning process but can not be categorized into policy learning, CABI[6] focuses on the model data augmentation, RAMBO[3] focuses on the mini-max optimization about the offline dataset. Even, as far as I know, the MPC methods only covers a relatively small portion of the model-based offline RL methods.

    Also, the setting of the paper seems to be more related to the so-called compositional/functional RL [12], which should be discussed and some of the baselines should be compared.

Minor issues in presentation:

1. What is $W$ in Proposition 4.2?

2. In Fig 2.c, shouldn't the expected outcome look like a diagonal matrix, since evaluating on the same policy where the embedding is trained (diagonal entry) should give the best performance?

[1] Model-Based Offline Reinforcement Learning with Pessimism-Modulated Dynamics Belief

[2] VALUE MEMORY GRAPH: A GRAPH-STRUCTURED WORLD MODEL FOR OFFLINE REINFORCEMENT LEARNING

[3] Robust Adversarial Model-Based Offline Reinforcement Learning

[4] Model-based Offline Reinforcement Learning with Count-based Conservatism

[5] EFFICIENT OFFLINE POLICY OPTIMIZATION WITH A LEARNED MODEL

[6] Double Check Your State Before Trusting It: Confidence-Aware Bidirectional Offline Model-Based Imagination

[7] A Unified Framework for Alternating Offline Model Training and Policy Learning

[8] CONSERVATIVE BAYESIAN MODEL-BASED VALUE EXPANSION FOR OFFLINE POLICY OPTIMIZATION

[9] Focal: Efficient fully-offline meta-reinforcement learning via distance metric learning and behavior regularization

[10] Fast adaptation to new environments via policy-dynamics value functions.

[11] Universal Value Function Approximators.

[12] A survey on continual learning and functional composition.

**Questions:**

See Weakness above.

---

> ### Author Response · Authors · 2023-11-20
> **official response (part 1)**
>
> Thank you for your insightful comments. we response the questions and concerns as follows:
> ### **Q1: Limited technical contributions. The central idea of the paper is training policy-conditioned dynamics models instead of a single model on a diverse offline RL dataset to achieve better modeling error. However, the idea of using MDP-conditioned embeddings for training a universal model in deep RL is not new, see for example [9][10][11].**
>
> While we appreciate the references to existing literature on MDP-conditioned methods in deep RL, we believe that the technical contributions of this paper is beyond the method implementation itself. The paper presents a novel perspective on leveraging MDP-conditioned dynamics models, particularly in the context of offline RL datasets.
>
> We would like to first highlight the distinct aspects of our approach.
>
> - The referenced works, such as Universal Successor Features Approximators [11], FOCAL [9], and Policy-Dynamics Value Functions [10], indeed employ MDP-conditioned methods. However, these approaches primarily focus on adapting to diverse environmental conditions and reward structures through modifications prior to policy application.
> - In contrast, our work diverges significantly in both intent and implementation. We utilize policy-conditioned methods not as a means to adapt to environmental variability, but rather to enable different policies to learn more effectively from a dynamic model tailored to the dataset's diversity. This approach specifically targets the reduction of value gaps, a critical aspect in offline RL that has not been the primary focus of the aforementioned studies.
>
> To the best of our knowledge, this particular application of policy-conditioned methods to enhance policy learning in the context of diverse offline RL datasets represents a novel contribution. It extends beyond the scope of existing methods by prioritizing the alignment between policies and the prediction focus of dynamic models, which is a departure from the primarily environment-focused adaptations in prior research.
>
> We believe this clarification underscores the unique technical contributions of our work and its significance in advancing the field of deep reinforcement learning.
>
> ### **Q2: It only shows that the generalization error can be bounded given a Lipshitz assumption, without explictly modeling the effect of policy embedding. The main argument seems to be that the adaptation gain of PCM is positive but zero for PAM.**
>
> The analysis is in fact an ideal case where the policy information could be perfectly injected into the model, under which case the generalization to other policies naturally relies on the Lipschitz continuity of the learned model.
>
> The policy embedding is a practical method to realize such the information injection mechanism, however, it is hard in general to analyze the effect of the concrete embedding, and we resort to the adaptation gain argument, which is an overall description of the model adaptation effect. In the previous submission, we discussed three possible mode of the adaptation in Appendix A.3, where the Case 2 Imaginary retraining is an ideal model of fine-tuning PAM for the target policy. However, it is just “imaginary” since it requires the real target domain data, which is inaccessible in our setting as well as many realistic situations, and we regard it as a ceiling of model accuracy for any practical PCM algorithms. If we cannot obtain the target policy data, the PAM can never be tuning specially for the target policy, and the adaptation gain is exactly zero. We believe the case 3 Intermediate adaptation reflects the adaptation effect of practical PCM methods, which lies between the two extreme cases of perfect adaptation ( case 2. imaginary retraining) and zero adaptation gain (case 1. direct match) and is able to partially reduce the extrapolation error. The empirical results in Sec 5.2 provides evidence to support this discussion.

---

> ### Author Response · Authors · 2023-11-20
> **official response (part 2)**
>
> ### **Q3: One can simply fine-tune PAM given the data of a new policy to achieve a positive adaptation gain, which is also a stronger baseline should be compared in experiments in 5.3.1.**
>
> We appreciate your suggestion to compare our PCM with a fine-tuned PAM as a baseline in our experiments. However, we would like to clarify a key practical limitation associated with fine-tuning PAM for new policies, particularly in an offline policy evaluation/learning setting.
>
> Fine-tuning PAM parameters for new policies, while theoretically feasible, is not practically viable in our context. This approach necessitates interactions of the new policies with the real environment to gather target domain experiences, which is prohibitive in offline settings where such interactions are not possible or desired. In contrast, our PCM is designed to adapt to new policies at runtime without the need for model parameter fine-tuning or real-world interaction experiences with the target policies. As mentioned above, we analysed this  in Appendix. For full discussion, please refer to case 2 (imaginary retraining) in Appendix A.3, which actually is the ideal adaptation method.
>
> Nevertheless, recognizing the theoretical value of fine-tuned PAM as an algorithm for comparison, we have included its performance in our revised paper to serve as an upper-bound comparison. This inclusion provides a perspective on the *optimal* adaptation gain achievable under less constrained conditions. The following table presents the value gaps for both PAM and fine-tuned PAM, compared with PCM:
>
> |  | PAM | PAM(fine-tuned) | PCM |
> | --- | --- | --- | --- |
> | halfcheetah-medium-replay | 0.49±0.09 | 0.25±0.07 | 0.31±0.08 |
> | walker2d-medium-replay | 0.38±0.07 | 0.18±0.06 | 0.25±0.05 |
>
> These results show that while there is a margin between PCM and the optimally fine-tuned PAM, PCM nonetheless achieves substantial adaptation gains compared with the original PAM. This demonstrates the practical efficacy and adaptability of PCM in offline policy evaluation/learning scenarios. Due to time limitations, we will add all results to other tasks later.
>
> We believe this additional comparison enriches our experimental analysis and provides a more comprehensive understanding of PCM's performance relative to other approaches.
>
> ### **Q4: I believe it's important to see whether PCM outperforms other competitive model-based OPE baselines when using the same architecture. A simple finetuned PAM on unseen task should also be compared as a stronger baseline (see point 2)**
>
> Thank you for your valuable feedback regarding the comparison of our PCM with other model-based and model-free OPE methods.
>
> Firstly, it is important to note that among the baselines considered in our study, PAM is the only model-based OPE method. The remaining baselines are model-free OPE methods. This distinction is crucial as it ensures a fair and relevant comparison within the context of model-based OPE approaches.
>
> Regarding the suggestion to compare our approach with a fine-tuned PAM on unseen tasks, we argue that a fine-tuned PAM should not be considered an additional baseline. This is due to its reliance on real interaction trajectories of the target policies, which diverges from the primary objective of OPE methods - to evaluate policies without necessitating further real-world interactions.
>
> Furthermore, to bolster our comparison and validate the efficacy of PCM, we have also implemented PCM with a standard Multilayer Perceptron (MLP) architecture. This implementation demonstrates that the PCM mechanism consistently aids in reducing value gaps across different architectures. The results of this implementation can be found in the general response.

---

> ### Author Response · Authors · 2023-11-20
> **official response (part 3)**
>
> ### **Q5: The review of related work of model-based offline RL is inadequate.**
>
> Thank you for your comments regarding the review of related work in our paper. Our approach to categorizing related work is based on how the models are utilized within the offlineRL framework.
>
> Specifically, we distinguish between methods that use models for model predictive control (MPC), where policies are generated through planning, and those used for policy learning (PL). This classification is designed to evaluate the effectiveness of models in facilitating downstream tasks, which is central to our study.
>
> Considering the recent works you mentioned, we acknowledge that they indeed present novel approaches within the model-based offline RL domain. For instance, AMPL [7] and RAMBO [3] integrate dynamic-model training with policy learning, and CABI [6] focuses on generating new datasets for offline RL policy learning in an adversarial manner. According to our classification principle, these methods primarily support policy learning as they leverage dynamics models for this purpose.
>
> We have chosen this classification scheme because it aligns closely with the objectives of our research and informs the design of our experiments. While we recognize other valid methods of classification, we believe our approach is more pertinent to the specific issues and questions our paper addresses. To this end, we have updated our manuscript to include additional related works in the main text, thereby providing a more comprehensive overview.
>
> Regarding your point about compositional/functional RL, we respectfully disagree with the applicability of this concept to our work. Compositional RL typically assumes that tasks can be broken down into subtasks, which is not an assumption we make in our approach. Our method focuses on learning representations about tasks without decomposing them into subfunctions, distinguishing it from compositional methods.
>
> We hope this response adequately addresses your concerns and provides clarity on our classification rationale and its relevance to our research objectives.
>
> ### **Q6: What is $W_1$ in Proposition 4.2?**
>
> We apologize for the omitted explanation for it. Here, $W_1$ is the Wasserstein-1 distance.
>
> ### **Q7: In Fig 2.c, shouldn't the expected outcome look like a diagonal matrix, since evaluating on the same policy where the embedding is trained (diagonal entry) should give the best performance?**
>
> The ideal result is that it appears as a diagonal matrix. However, due to some randomness in experiments, it’s hard to get an exact, “perfect” diagonal matrix. The result shown in Fig 2.c is close to a diagonal matrix, in which we can see that the model error near the diagonal is small while the error away from the diagonal is large.

---

### Author Response · Authors · 2023-11-20
**general response**

We appreciate the reviewers for their thoughtful and constructive comments, which have been instrumental in enhancing the quality of our manuscript. We are encouraged by their recognition of several key aspects of our work:

1. **Novel Approach in Model Learning for RL:** Reviewers Ag2q, ENii, and BstF have noted our approach in reducing modeling error in MDP dynamics through policy-conditioned models. This method aligns with current trends in context-based meta-learning, as highlighted by Ag2q. ENii and BstF also acknowledged the relevance and potential impact of this approach in real-world scenarios.
2. **Clarity and Structure of the Paper:** The reviewers commended the clear and organized presentation of our paper. This acknowledgment of the paper's readability and structure, especially in the methodology section, is encouraging (BstF, ENii).
3. **Empirical Validation of Proposed Solution:** The practical effectiveness of our Policy-Conditioned Model (PCM) learning is supported by our experimental results. Reviewers noted its performance advantages in various tasks, particularly in robotic tasks (BstF, ENii), indicating the method's applicability.
4. **Theoretical Justification and Intuitive Approach:** Our work's theoretical aspects were recognized for supporting the intuition behind our algorithm (Reviewer BstF). We believe this combination of theoretical grounding and intuitive design contributes to a better understanding of our approach.

We have incorporated the reviewers' suggestions by adding extra explanations, experiments, and discussions on additional related works. All the revision are colored with purple. The significant revisions are summarized as follows:

- **Providing some practical scenarios that the multi-source nature of offline datasets is a common phenomenon**: In introduction, we give some example that the multi-source nature of offline datasets is a common phenomenon, according to the reviewer ENii;
- **Providing a formal definition of “model learning on diverse datasets”**: We have provided a formal definition of “model learning on diverse datasets” as well as some remarks in Section 4.1 according to the reviewer ENii, which help readers have a better understanding of this setting.
- **Enriching the related work**:  We have discussed more work in the related work section, including some recent work on OPE and MB offline RL, according to reviewers Ag2q and U3YJ.
- **Conducting more experiments**:  We have additionally evaluated baselines as well as our method on 5 more tasks for OPE tasks, evaluated two more methods (Importance Sampling and DICE) for OPS tasks, PCM with the MLP architecture, PAM-fintune as a upper-bound algorithm with optimal adaptation gain, and provided visualizations to learned embeddings on more environments, according to reviewers ENii, BstF, U3YJ and Ag2q.

---

> ### Author Response · Authors · 2023-11-20
> **Additional Expeirments**
>
> ### Evaluate PCM in more datasets
>
> We have supplemented five datasets for OPE task, including ant-medium-replay, ant-medium-expert, halfcheetah-medium-expert, hopper-medium-expert, and walker2d-medium-expert. Similarly, our method also outperforms other methods significantly on these tasks. We update the averaged scores in Absolute error, Rank correlation, and Regret in Figure 4.  Please see Table 3 to Table 6 for raw results in our revised paper.
>
> As can be seen in Figure 4, our method, PCM, consistently outperforms the main baseline, PAM, and other methods in the supplementary experiments across **11** datasets. This is evidenced by PCM achieving the lowest absolute error, the highest rank correlation, and the least regret, indicating superior accuracy, consistency in policy ranking, and closer performance to the optimal policy within Off-Policy Evaluation tasks.
>
> ### Ablation study: PCM in MLP architecture
>
> To bolster our comparison and validate the efficacy of PCM, we have also implemented PCM with a standard Multilayer Perceptron (MLP) architecture. This implementation demonstrates that the PCM mechanism consistently aids in reducing value gaps across different architectures.
>
> |  | FQE | DR | IS | DICE | VPM | PAM (old arch) | PCM (old arch) |
> | --- | --- | --- | --- | --- | --- | --- | --- |
> | hopper-medium-replay | 0.43 | 0.49 | 0.69 | 0.67 | 0.72 | 0.31 | **0.18** |
> | walker2d-medium-replay | 0.56 | 0.53 | 0.76 | 0.67 | 0.76 | 0.43 | **0.34** |
> | halfcheetah-medium-replay | 0.50 | 0.50 | 0.75 | 0.72 | 0.69 | 0.56 | **0.44** |
>
> The experimental results presented in the table indicate a consistent improvement when applying PCM as opposed to PAM across multiple environments. For example, in the hopper-medium-replay dataset, PCM achieved a significant reduction in the value gap, dropping to 0.18 compared to PAM's 0.31, showcasing a 41.9% improvement. In walker2d-medium-replay and halfcheetah-medium-replay, PCM also outperformed PAM with a decrease in the value gap of 20.9% and 21.4%, respectively. These results suggest that the PCM mechanism enhances the accuracy of value estimation across various architectures, including MLP, and confirms the robustness and generalizability of PCM in model learning.
>
> ### Offline Policy Selection with more baselines
>
> We have additionally evaluated IS and DICE for off-policy selection. The results of OPS are:
>
> | **Task Name**              | **Last Epoch** | **FQE** | **IS**  | **DICE** | **PAM** | **PCM** |
> |----------------------------|----------------|---------|---------|----------|---------|----------------|
> | halfcheetah-medium-replay  | 39.3%          | 23.0%   | 87.8%   | 1.6%     | 1.6%    | **98.4%**      |
> | hopper-medium-replay       | 56.0%          | 34.1%   | 56.0%   | 19.8%    | 47.3%   | **64.8%**      |
> | walker2d-medium-replay     | -4.6%          | 4.6%    | 34.3%   | 13.0%    | -30.6%  | **51.9%**      |
> | **Average**                | 30.2%          | 20.6%   | 59.4%   | 39.3%    | 11.5%   | **71.7%**      |
>
>
> We observe that IS indeed has more competitive performance (59.4% performance gain) than FQE (20.6% performance gain) but still falls behind PCM (71.7% performance gain).  The detailed results can be found  in Page 23 Table 8-9 in the appendix, and Table 2 in the main body
>
> #### fintune of PAM
> As suggested by Ag2q, PAM with fine tuning in target policies's datasets can be regarded as the upper-bound algorithm with the optimal adaptation gain. We implement PAM with finetuning in two datasets. We will add the results in more tasks if completed.
>
> |  | PAM | PAM  (fine-tuned) | PCM |
> | --- | --- | --- | --- |
> | halfcheetah-medium-replay | 0.49±0.09 | 0.25±0.07 | 0.31±0.08 |
> | walker2d-medium-replay | 0.38±0.07 | 0.18±0.06 | 0.25±0.05 |
>
> These results show that while there is a margin between PCM and the optimally fine-tuned PAM, PCM nonetheless achieves substantial adaptation gains compared with the original PAM. This demonstrates the practical efficacy and adaptability of PCM in offline policy evaluation/learning scenarios

---

### Meta-Review · Area_Chair_2bbN · 2023-12-03

**Metareview:**

The paper proposes a new technique for offline model based RL where the model learnt consists of a mixture of models conditioned on different policies.  This allows an agent to learn more accurate models conditioned on the policy.  However, as pointed out by the reviewers, this is a relatively mild contribution that resembles or inspires itself from related works.  Also, similar gains in accuracy may be achieved by fine-tuning a policy agnostic model.  While the rebuttal helped to address many concerns, the contributions lack significance and the empirical results are not as comprehensive as one might expect given the empirical nature of the work.

**Justification For Why Not Higher Score:**

The continutions lack significance and the empirical results are not comprehensive.

**Justification For Why Not Lower Score:**

N/A

---

### Decision · Program_Chairs · 2024-01-16

Reject